# Identification of Stage-Specific microRNAs that Govern the Early Stages of Sequential Oral Oncogenesis by Strategically Bridging Human Genetics with Epigenetics and Utilizing an Animal Model

**DOI:** 10.3390/ijms25147642

**Published:** 2024-07-12

**Authors:** Iphigenia Gintoni, Stavros Vassiliou, George P. Chrousos, Christos Yapijakis

**Affiliations:** 1Unit of Orofacial Genetics, 1st Department of Pediatrics, School of Medicine, National Kapodistrian University of Athens, “Aghia Sophia” Children’s Hospital, 115 27 Athens, Greece; iph.gintoni@gmail.com; 2Department of Molecular Genetics, Cephalogenetics Center, 176 72 Athens, Greece; 3Department of Oral and Maxillofacial Surgery, School of Medicine, National Kapodistrian University of Athens, Attikon Hospital, 124 62 Athens, Greece; stvasil@med.uoa.gr; 4University Research Institute for the Study of Genetic and Malignant Disorders in Childhood, Choremion Laboratory, “Aghia Sophia” Children’s Hospital, 115 27 Athens, Greece

**Keywords:** oral squamous cell carcinoma, OSCC, oral cancer, miRNA, miRNA expression, genes, gene expression, hyperplasia, dysplasia, early invasion, early stage OSCC, in situ carcinoma, precancerous lesions, animal model, early diagnosis, liquid biopsy

## Abstract

Oral squamous cell carcinoma (OSCC) is a highly prevalent and aggressive malignancy, with mortality rates reaching 60%, mainly due to its excessive diagnostic delay. MiRNAs, a class of crucial epigenetic gene-expression regulators, have emerged as potential diagnostic biomarkers, with >200 molecules exhibiting expressional dysregulation in OSCC. We had previously established an in silico methodology for the identification of the most disease-specific molecules by bridging genetics and epigenetics. Here, we identified the stage-specific miRNAs that govern the asymptomatic early stages of oral tumorigenesis by exploiting seed-matching and the reverse interplay between miRNA levels and their target genes’ expression. Incorporating gene-expression data from our group’s experimental hamster model of sequential oral oncogenesis, we bioinformatically detected the miRNAs that simultaneously target/regulate >75% of the genes that are characteristically upregulated or downregulated in the consecutive stages of hyperplasia, dysplasia, and early invasion, while exhibiting the opposite expressional dysregulation in OSCC-derived tissue and/or saliva specimens. We found that all stages share the downregulation of miR-34a-5p, miR124-3p, and miR-125b-5p, while miR-1-3p is under-expressed in dysplasia and early invasion. The malignant early-invasion stage is distinguished by the downregulation of miR-147a and the overexpression of miR-155-5p, miR-423-3p, and miR-34a-5p. The identification of stage-specific miRNAs may facilitate their utilization as biomarkers for presymptomatic OSCC diagnosis.

## 1. Introduction

Oral cancer, encompassing areas such as the lips, gums, tongue, and palate, is one of the most frequently occurring malignancies, ranking sixth in terms of frequency around the globe [1,2,3]. More than 90% of oral cancer cases are histologically represented by oral squamous cell carcinoma (OSCC), which originates from the stratified squamous cell epithelial lining of the oral cavity, where it can develop from clinically normal mucosa or premalignant lesions [4,5]. OSCC is considered a highly prevalent and lethal malignancy due to its highly aggressive behavior, marked by invasive growth and a bold propensity for metastasis. In 2023, 377.713 new OSCC cases, as well as 177.757 deaths were reported globally [4,6]. Despite advancements in the field of treatment, the prognosis for OSCC remains disheartening, with a five-year survival rate of about 40% [2,6,7,8]. At the same time, following initial treatment, one in two patients exhibits recurrence, metastasis, or second primary malignant tumor development, mainly within the first two years [1,4].

One of the major challenges contributing to the high mortality rates associated with OSCC is late diagnosis. In fact, around 50% of patients are diagnosed in advanced stages (III or IV), when lymphatic infiltration has already occurred [1,7,8]. This is mainly because the early stages of oral carcinogenesis are usually asymptomatic, hence, the vast majority of patients delay in seeking dental or medical attention until persistent symptoms occur, which histologically translates to an already formed, aggressive neoplasm [7]. Moreover, this can also occur due to a misdiagnosis or negligence of a possibly precancerous oral condition such as leukoplakia or erythroplakia, which are common predecessors of OSCC and are occasionally found adjacent to malignant lesions [2,5,7]. Current diagnostic approaches, which involve a clinical oral examination, followed by a biopsy of the suspicious tissue, as well as MRI and CT scans, can be excessively time-consuming, also adding to the already existing problem of a “ticking clock” [2,4]. The importance of developing reliable diagnostic techniques to detect presymptomatic OSCC in its early stages is underlined, considering that timely diagnosis in stages I or II can more than double the survival rates of patients up to 80% [1,7,9].

Minimally or non-invasive methods, such as “liquid biopsy”, which involve the screening of specific molecules in biofluids as diagnostic markers, have demonstrated substantial promise for clinical application in the field of precision oncology, regarding cancer detection in manageable early stages [10,11,12,13]. OSCC stands out as an ideal neoplasm for liquid biopsy, primarily due to its accessibility and its direct interaction with the oral environment, ensuring that the molecules of interest, such as microRNAs, exosomes, cell-free tumor DNA, or circulating tumor cells, are shed directly from the tumor site into the saliva, making it a particularly valuable sample for oral cancer detection, even in the earliest of stages. However, reliable detection primarily requires the identification of the most specific and representative disease-reflective biomarkers, a task that remains challenging [9,11,14].

MicroRNAs (miRNAs) have emerged as key players in liquid biopsy, as they offer several advantages as biomarkers, including stability in body fluids, as well as characteristic expressional dysregulation patterns (significant up- or downregulation) in cancerous compared to normal tissues [8,9]. MiRNAs are small, single-stranded RNA molecules of 18–25 nucleotides, that regulate gene expression by complimentarily binding to specific sites within the untranslated regions (UTRs) of the messenger RNAs (mRNAs) of their target-genes. This miRNA/mRNA interaction leads to mRNA degradation or inhibition of protein translation, thereby reducing the levels of the encoded protein products [3,8,12,15,16].

In recent years, numerous miRNAs have been extensively studied for their expression patterns in OSCC, reflecting their potential as diagnostic targets [6,8,9,17,18,19,20,21,22,23,24,25,26,27,28]. However, this surge in research has led to a plethora of data, with over 200 miRNAs reported to be either upregulated or downregulated in OSCC-derived samples (tissue, saliva, and blood), while the majority exhibit similar dysregulation patterns in other malignancies as well. Therefore, significant challenges occur as to the interpretation of experimental data, but mainly the identification of the few truly disease-specific molecules that can be reliably utilized as biomarkers for OSCC diagnosis, especially in its early asymptomatic stages [8].

Recently, in response to the above challenge, our group developed a combinatory bioinformatic methodology, which managed to identify the five most OSCC-specific miRNAs (miR-34a-5p, miR-155-5p, miR-124-3p, miR-1-3p, and miR-16-5p), out of all experimentally validated miRNAs that exhibit dysregulation in OSCC and are proposed as potential diagnostic markers [8]. This became possible through exploiting the intricate relations between genetics and epigenetics and the expressional reverse interplay between miRNA and target-gene expression levels. This in silico methodology was designed to identify the most disease-specific miRNA molecules involved in the development of OSCC, based on (a) their ability to simultaneously target and possibly regulate at least 60% of the 15 most important OSCC-related oncogenes and more than 60% of the five mainly implicated tumor suppressor genes that exhibit characteristic upregulation and downregulation in OSCC tumors, respectively [8,18,29], and (b) the nature of their reported dysregulation (up- or downregulation) in OSCC-derived specimens, to functionally document their putative regulatory role in OSCC development, through the inverse relations between miRNA levels and the expression of their target genes [8].

Building upon our prior research on OSCC, we utilized experimental epigenomic data on miRNA expression in humans and genomic expression data constituting the genetic signature of each early OSCC stage, spanning from precancerous hyperplasia and dysplasia to the early invasion of malignant OSCC cells. Our genetic data were derived from the findings of the award-winning experimental hamster model of sequential oral oncogenesis, previously developed by our group [30,31]. Although the involved genes have been strongly associated with human OSCC, such a staging system of gene expression during each consecutive stage of oral oncogenesis is not yet available in humans, while this animal model is considered to be one of the most reflective of the respective human pathology. This animal system encompasses all of the distinctive stages of oral carcinogenesis, ranging from the earliest stage of precancerous hyperkeratosis to moderately developed OSCC, illustrating the signature expressional patterns of critical oncogenes and tumor suppressor genes at each stage [30,31]. The aim of this study lies in identifying the most stage-specific miRNAs that govern the consecutive initial histological phases of OSCC tumorigenesis. More specifically, the progression from hyperplasia to precancerous dysplasia, and ultimately the malignant transformation from dysplasia to early invasion of OSCC cells, which also incorporates the “stage 0” of “in situ carcinoma”, the timely detection of which could drastically impact prognostic and therapeutic outcomes for patients with OSCC.

## 2. Results

The aim of the present study was to identify the most stage-specific miRNA molecules that govern the consecutive initial histological stages of OSCC tumorigenesis through their dysregulation. Those include the transition from hyperplasia to precancerous dysplasia, and the subsequent malignant transformation from dysplasia to early invasion of OSCC cells. To accomplish this objective, a sequence of consecutive in silico processes and bioinformatic analyses for each selected stage were employed. The results of each phase of the study are comprehensively described below.

### 2.1. Databases of Dysregulated miRNA Molecules in OSCC Tissue/Saliva Specimens

The two databases, which were developed during previous bioinformatic research on OSCC-specific miRNA molecules for OSCC, encompassed a total of 239 dysregulated molecules, including 106 significantly upregulated and 133 downregulated miRNAs that have been reported in the OSCC-related literature up until 20 June 2023 [8]. In this study, a supplementary search was conducted by employing the same criteria to provide updated results for the period between 21 June 2023 and 10 April 2024. For the purposes of this research, we furtherly confined our search to miRNA quantification results originating from tissue and saliva specimens, in order to increase the specificity of our results. After compiling miRNA expression data from each of the five reviews that met our search criteria, 7 miRNA molecules that were upregulated and 10 miRNA molecules that were downregulated in OSCC tissue and/or saliva (Table 1), and were not included in our previously developed respective databases, were yielded.

### 2.2. Stage-Specific miRNAs Governing Oral Hyperplasia

The bioinformatic strategy employed to forecast the miRNAs that target at least one of the eight genes in the developed upregulated gene panel for the precancerous stage of oral hyperplasia resulted in the identification of 647 putative miRNA molecules (Figure 1). From this set of results, 12 miRNAs were predicted to simultaneously target the mRNAs of over 75% of the target genes within the developed panel. This suggests that these miRNAs may target and potentially regulate more than six out of the eight upregulated target-genes. Finally, according to the last filtering step, only 3 out of the latest set of 12 miRNAs exhibit significant downregulation in OSCC-derived biosamples. More specifically, miR-34a-5p, which targets all eight (100%) (Figure 2), while exhibiting significant downregulation in OSCC tissue and saliva specimens, miR-124-3p with a target score of seven out of eight upregulated genes (Figure 3) and significantly decreased levels in OSCC tissue and saliva samples, and, finally, miR-125b-5p that targets seven out of eight genes as well (Figure 4), and exhibits downregulation in OSCC tumor tissue, are yielded as the most stage-specific for the precancerous stage oral hyperplasia (Table 2).

### 2.3. Stage-Specific miRNAs Governing Oral Dysplasia

The bioinformatic approach employed to predict the miRNAs that target at least one of the nine genes in the upregulated gene panel for the following precancerous stage of oral dysplasia, yielded 658 potential miRNA molecules (Figure 5). From this set of results, five miRNAs simultaneously were predicted to partake in complementary binding sites on the mRNAs of over 75% of the gene targets within the developed panel. This suggests that these miRNAs might target and potentially regulate at least seven out of the nine upregulated target genes in oral dysplasia. Finally, according to the last filtering step, four out of the latest set of five miRNAs exhibit significant downregulation in OSCC-derived biosamples. More specifically, miR-34a-5p, which targets all nine (100%) (Figure 6) and exhibits significant downregulation in OSCC tissue and saliva specimens, miR-124-3p with a target score of seven out of nine and significantly downregulated expression in OSCC-derived tumor and saliva specimens (Figure 7), miR-125b-5p that targets seven out of nine genes, and demonstrates downregulation in OSCC tissue (Figure 8), and, finally, miR-1-3p with a target score of seven out of nine upregulated genes and significantly decreased levels in OSCC tissue specimens (Figure 9), are yielded as the most stage-specific for the precancerous stage oral dysplasia (Table 3).

As to the downregulated gene panel consisting of three downregulated genes in oral dysplasia, the target prediction analysis reveals 276 miRNA molecules that target each one of the three downregulated target genes (Figure 10). Out of 276, only 3 candidate miRNAs that target and possibly regulate all (100%) of the developed panel’s genes at the same time, were identified. However, none of these miRNAs passed the final filtering step because they have not been reported as upregulated in OSCC tissue or saliva specimens.

### 2.4. Stage-Specific miRNAs Governing OSCC Early Invasion

Regarding the initial malignant stage of early invasion, 303 miRNA molecules were predicted to target at least one of those five genes that comprised the upregulated gene panel for this particular histological phase of tumorigenesis (Figure 11). From this set of results, only six miRNAs were predicted to simultaneously hold complementary binding sites on the mRNAs of over 75% of the gene targets within the developed panel, suggesting that they might target and possibly regulate the post-transcriptional expression of at least four out of the five genes.

Finally, according to the last filtering step of this strategy, five out of this total of six miRNAs exhibit significant downregulation in OSCC-derived tissue and/or saliva samples. In particular, miR-34a-5p, which demonstrates an ultimate target score of five out of five genes (100%) (Figure 12), while exhibiting significant downregulation in OSCC tissue and saliva specimens, followed by miR-124-3p (Figure 13), miR-125b-5p (Figure 14), miR-1-3p (Figure 15), and miR-147a (Figure 16) with target scores of four of five upregulated genes in different combinations, while exhibiting significantly decreased expression levels in OSCC tumor tissue and/or saliva samples. Hence, those five molecules are designated as the most stage-specific for the histological stage of malignant OSCC early invasion (Table 4).

In respect to the downregulated gene panel for this stage, the *CDKN2A* and *BCL2* genes, with opposing roles in OSCC, but with the same stage-specific expression pattern, according to the hamster model, are involved. The employed target prediction analysis reveals 159 miRNA molecules (Figure 17), targeting at least one of the two genes and a set of 10 miRNAs that simultaneously target more than 75%, which translates as two of two downregulated genes (100% of the panel). From those 10 molecules, only 3 are experimentally reported to exhibit significant upregulation in OSCC-derived tissue and/or saliva samples (Table 4). Those three ultimate results, serving as the most stage-specific miRNAs governing OSCC early invasion, include miR-155-5p (Figure 18), which is characteristically upregulated in both tissue and saliva samples [8,26] and miR-423-3p (Figure 19), which scores characteristically high levels in OSCC tumor specimens, as well as miR-34a-5p (Figure 20) that has been reported as upregulated in saliva of OSCC patients according to a minimal number of studies.

## 3. Discussion

Despite some advancements in diagnosis and treatment, the outlook for OSCC remains bleak, as indicated by its approximately 40% five-year survival rate [2,6,7,8], while recurrence or metastasis is extremely likely to develop within the first two years post-treatment [1,4]. Late diagnosis is one of the main obstacles that contribute to the high mortality rates associated with OSCC, with one in two patients getting diagnosed in an already advanced stage [1,7,8]. MiRNAs, a class of crucial epigenetic regulators of gene expression, have emerged as key players in precision oncology, serving as ideal biomarkers for liquid biopsy, since they exhibit strong diagnostic potential by reflecting a tissue’s malignant state through the quantification of their expressional dysregulation [3,8,9,12,16]. In recent years, the expression of numerous miRNAs has been studied in OSCC, resulting in a vast number of experimental data with more than 250 molecules reported to be either upregulated or downregulated in OSCC-derived samples (tissue, saliva or blood) [6,8,9,17,18,19,20,21,22,23,24,25,26,27,28], while the majority exhibits similar dysregulation in other neoplasms as well. Consequently, there is a state of disarray regarding the interpretation of findings and the identification of the most representative molecules for the disease at large, but particularly for its early asymptomatic stages [8].

In 2023, our group established a combinatory bioinformatic methodology that managed to identify the five most specific miRNAs for OSCC (miR-34a-5p, miR-155-5p, miR-124-3p, miR-1-3p, and miR-16-5p) out of all miRNAs that exhibit dysregulation in the malignancy, thus serving as the most suitable of all proposed miRNA biomarkers for minimally invasive biopsy. This became possible by exploiting the intricate relationship between genetics and epigenetics, as well as the expressional reverse interplay that exists between the levels of miRNA molecules and those of their target-genes [8]. However, the challenge of revealing highly specific biomarkers that can reflect the carcinogenic potential of a precancerous lesion or even a seemingly healthy oral mucosa that already hosts dysplastic or invasive malignant OSCC cells, remained unsolved.

Continuing and advancing this previous work, we decided to dive deeper into the epigenetic mechanisms of OSCC, by identifying the most important miRNA molecules that govern the distinct initial stages of oral tumorigenesis that precede the development of a clinically detectible malignant lesion. Those include the precancerous stages of oral hyperplasia and dysplasia and the malignant stage of early invasion, which are typically asymptomatic, yet critical for timely therapeutic intervention that could hold the key to improving the prognosis of patients, by almost doubling their odds for survival. To identify the miRNAs in question, we employed our previously developed methodology (for the identification of disease-specific molecules) and tailored it in a stage-specific setup, by inputting genomic data, highly reflective of each early OSCC histological stage. These data were derived from our hamster model of sequential oral oncogenesis [30,31]. This animal system is quite representative of human OSCC and encompasses the expressional variations of critical cell-cycle regulatory genes through the course of oral tumorigenesis, thus portraying the genetic expressional signature of each stage spanning from clinically normal oral mucosa to moderately differentiated OSCC tumors. We focused on the precancerous stages of hyperplasia and dysplasia, as well the initial malignant stage of early invasion, in an attempt to capture malignant transformation.

Following the updating of our developed databases on experimentally verified dysregulated miRNAs in OSCC, we developed two gene panels for each stage of OSCC oncogenesis (hyperplasia, dysplasia, early invasion), according to their expression (up- or downregulation) during each stage, according to this established animal model. A primary bioinformatic analysis of miRNA/target interactions was performed on each panel at each stage to identify all miRNAs that are expected to target and, thus, potentially regulate the expression of each included gene. This step was followed by a filtering process, during which miRNAs that simultaneously target over 75% of each panel’s genes were extracted. Finally, from those sets of results, the miRNA molecules that exhibit the opposite expressional dysregulation in OSCC-derived tissue and/or saliva specimens, with respect to one of their target genes, were yielded as the most stage-specific for each of the histological stages. In this study, we particularly confined our filtration criteria to miRNA molecules that exhibit dysregulation in tissue and saliva samples, to enhance the specificity and reliability of our results, by increasing the possibility that the miRNAs in question are from the tumor itself or are directly secreted into the surrounding saliva from the malignant lesion.

According to our findings, the most stage-specific dysregulated miRNAs for oral hyperplasia are the significantly downregulated molecules miR-34a-5p, miR-124-3p (tissue and saliva), and miR-125b-5p (tissue). As to the next stage of oral dysplasia, miR-34a-5p, miR-124-3p, and miR-125b-5p that exhibit diminished expression in OSCC-derived tissue and saliva samples, and, finally, miR-1-3p, which demonstrates significant downregulation in OSCC tumor tissue, are yielded as the most stage-representative. Lastly, for the initial cancerous phase of early invasion, the downregulated miR-34a-5p, miR-124-3p (tissue and saliva), miR-125b-5p, miR-1-3p (tissue), and miR-147a (saliva) are the most stage-specific and indicative of malignant transformation.

Conclusively, the stages of oral hyperplasia, dysplasia, and OSCC early invasion are characterized by the low expression of three common microRNAs (miR-34a-5p, miR124-3p, and miR-125b-5p). The dysplasia stage is distinguished from the preceding hyperplasia by the presence of the additional under-expressed miR-1-3p, which also serves as a linkage between the precancerous stage of dysplasia and the cancerous early invasion of OSCC cells. Lastly, the initial malignant stage of early invasion is entirely distinguished from the preceding two precancerous stages, by means of three additional molecules: miR-147a, the expression of which is considerably diminished, as well as miR-155-5p and miR-423-3p, whose expression is substantially increased in OSCC, but also miR-34a-5p, which has been found to be overexpressed in a small number of studies, unlike the majority, which reports it as significantly downregulated in this particular malignancy (Table 5).

The key role of miR-34a-5p in OSCC remained a big mystery in our previous work as well, as it met the criteria for yielding the most disease-specific molecule in the upregulated oncogene panel with a target score of 100% (15/15 oncogenes), as well as within the downregulated tumor suppressor gene panel by targeting 5/5 genes, alongside miR-155-5p. Therefore, its stage-specific role that was revealed in this current study might provide insight into its multifaceted role in malignancy, granted that it is downregulated in the precancerous stages of hyperplasia and dysplasia, while the spike in its expression is detected during the initial malignant stage of early invasion. Hence, the fact that it is reported as downregulated in the majority of studies, but upregulated in a small subgroup, might rely on the presence of all histological stages in the OSCC tissue specimens from which RNA was extracted for its quantification. Thus, it is possible that the studies reporting its upregulation in OSCC included smaller-sized, recently developed tumors with a significant presence of the “early invasion” stage in their histological microenvironment.

Ultimately, considering the outcomes derived from our prior research [8] on identifying the five most disease-specific miRNAs (miR-155-5p, miR-34a-5p, miR-124-3p, miR-1-3p, and miR-16-5p) for overall OSCC (among 239 dysregulated molecules), it is evident from the current analysis that most of these disease-specific molecules appear at different stages of OSCC oncogenesis. Specifically, miR-34a-5p and miR-124-3p (downregulated) are found to coexist in all three studied stages. Disease-specific miR-1-3p is introduced during the stages of dysplasia and early invasion and finally, the most important upregulated disease-specific molecule, miR-155-5p, stands among the four dysregulated miRNAs that distinguish the cancerous from the precancerous state of oral mucosa (Table 5).

The sole constraint to the conduction of a study that incorporates human and animal molecular data, is the possible slight distinction between an animal model and the molecular makeup of the respective human system. In order to eliminate this limitation, we aimed to proactively address it during the designing phase of the study protocol, by utilizing this particular model of oral oncogenesis over any alternative animal system. The Syrian golden hamster (*Mesocricetus auratus*) cheek-pouch oral carcinogenesis model is widely recognized as the most well-established system, in which the events that lead to the development of malignant and premalignant oral pathologies, are of great resemblance to those governing the respective human pathologies. Furthermore, the genes that comprise each stage’s dysregulated expressional signature have already been confirmed to play a key part in the development and progression of human OSCC as well [8,18,29,32]. However, a definitive experimental staging system of their precise expressional disequilibrium, at each stage of human oral oncogenesis, is currently unavailable. Finally, all the experimental miRNA expression data incorporated for the analyses of this study are derived solely from research conducted on OSCC tissue and saliva specimens of human origin, while their reported dysregulation has already been strongly associated with OSCC pathogenesis.

Overall, this study has yielded valuable insights into the role of miRNAs in the early stages of oral oncogenesis, through mapping the genetic/epigenetic interactions that govern each stage, by leveraging on the complex interplay between miRNAs and their target genes. The approach of this current research is highly innovative, since although over 250 dysregulated miRNAs have been experimentally detected in OSCC and are available in the literature, the RNA extraction preceding their tissual quantification has been performed in OSCC specimens that normally contain all preceding histological stages within the same tissue sample. Therefore, their stage-specific role has been nearly impossible to be directly portrayed in the experimental laboratory setting and has remained unseen until now.

Armed with a comprehensive bioinformatic methodology that integrates experimental epigenomic and genomic data, we successfully identified these three sets of molecules as the most specific for the earliest stages of OSCC development, among a multitude of candidates. Amongst our findings, the miRNAs marking the transition from precancerous dysplasia to early invasion of malignant OSCC cells are considered of the highest importance, since their dysregulation governs malignant transformation, which includes the “stage 0” of the neoplasm. The revealing of both their disease-specific and stage-specific roles, combined with their stability and measurable expression in accessible biological specimens, elucidates their immense potential as diagnostic biomarkers that could potentially indicate the histological stage of oral mucosa in real time. Their prospective clinical application expands beyond diagnosing a visible, oral lesion of concern, but might also hold the key to presymptomatic detection of OSCC, during its earliest stages as a part of routine oral screening (Figure 21).

The presented targeted molecular profiling approach boldly underlines the potential of miRNAs as valuable biomarkers. In addition, the employed methodology might also serve as a crucial bridge between the global imperative to advance liquid biopsy and its actual implementation, by circumventing the noise generated by the multitude of dysregulated molecules, thus elucidating the select ones holding the highest diagnostic value.

## 4. Materials and Methods

The present study involves the integration of human miRNA expression data from OSCC tissue and/or saliva samples with stage-specific gene expression data from a highly representative animal model for sequential oral oncogenesis, developed by our research team [30,31]. The incorporated epigenomic and genomic data were integrated by utilizing a combinatory in silico methodology that was developed by our group in 2023, for the identification of the 5 most disease-specific miRNAs for OSCC, among 239 dysregulated molecules, by strategically bridging genetics with miRNA—epigenetics [8]. Both pieces of research are briefly reviewed prior to the comprehensive presentation of the methods applied in the current study.

### 4.1. OSCC-Specific miRNA Identification Strategy by Bridging Genetics and Epigenetics

The identification of the most significant and indicative miRNA molecules for OSCC was determined based on the combination of the following variables: (I) Their capacity to simultaneously target and potentially regulate multiple crucial OSCC genes, and (II) their experimentally verified dysregulation pattern (either up- or downregulation) in OSCC, which shall be reverse to the one of their target genes [8].

Following the creation of two databases, which contain a total of 239 miRNAs that have been reported to be upregulated or downregulated in the OSCC-related literature (up until 20 June 2023), two customized gene panels were generated based on their role and expression in the malignancy. The first panel included the fifteen primarily involved oncogenes that are significantly overexpressed in OSCC. The second panel comprised the five mainly implicated tumor suppressor genes that demonstrate significant downregulation in OSCC biospecimens. The genes included in each panel served as possible targets for miRNA regulation. In the initial stage of our methodology, we employed target prediction analyses to identify all miRNA molecules that are anticipated to target and potentially regulate each gene in each panel, resulting in a substantial number of molecules. After applying further filtering to the acquired data, we identified a particular subset of miRNAs that are expected to target at least 60% of the fifteen oncogenes and over 60% of the five tumor suppressor genes included in the generated panels. Ultimately, the molecules that successfully passed through the previous filtering processes for each panel, but also demonstrated a reverse dysregulation pattern in OSCC compared to their target genes, were designated as the most OSCC-specific miRNAs [8].

### 4.2. The Hamster Model of Sequential Oral Oncogenesis

In order to genetically map the step-by-step tumorigenesis of OSCC in hamsters, we have created an experimental system using golden Syrian hamsters (*Mesocricetus auratus*) (30,31). Oral carcinogenesis was induced in the involved animals by topically applying DMBA (9,10-dimethyl-1,2-benzanthracene), dissolved in paraffin oil, at a concentration of 0.5%. Weekly examinations were performed on the pouches of all animals to monitor tumor growth on the inner lining of the buccal mucosa. After a period of 10 weeks following carcinogen application, the oral pouches that were treated were removed from the sacrificed animals and subsequently subjected to pathological assessment and immunohistochemical staining with fluorescent antibodies.

This animal system includes the following stages of oral oncogenesis: normal mucosa, precancerous hyperkeratosis, hyperplasia, and dysplasia, as well as malignant early invasion, well-differentiated, and moderately differentiated OSCC. The expression of multiple factors, previously associated with human OSCC pathogenesis, was quantified during each of the sequential stages within the excised animal tissue specimens. More specifically, the expression levels of the *EGFR*, *ERBB2*, *ERBB3*, *FGFR2*, *FGFR3*, *MYC*, *NRAS*, *ETS1*, *HRAS*, *C-FOS*, *JUN*, and *MKI67* oncogenes, as well as the apoptosis markers *BAX* and *BCL2*, and, finally, the tumor suppressor genes *TP53* and *CDKN2A* (*P16)*, were assessed in each stage in order to depict their respective dysregulated genetic signature (Figure 22). The expression levels of their encoded proteins were quantified and compared between precancerous stages (hyperkeratosis, hyperplasia, dysplasia) and normal oral mucosa. Similarly, the expressional data derived from the malignant stages (early invasion, well-differentiated OSCC, and moderately differentiated OSCC) were compared to their respective median expression of the under-study variables in non-cancerous and precancerous stages [30,31].

It is crucial to emphasize that the listed genes have been confirmed to play a key part in the development and progression of human OSCC. This is due to the fact that their dysregulation disrupts multiple signaling pathways that are responsible for cell-cycle regulation, thus leading to oral carcinogenesis [8,18,29,32]. However, due to the absence of a definitive staging system, comprehending their precise behavior at each stage in the context of expressional disequilibrium is quite challenging.

### 4.3. Databases of Dysregulated miRNA Molecules in OSCC

Our previous bioinformatic research on disease-specific miRNA molecules for OSCC in general encompassed the creation of two databases, including the total of significantly upregulated and downregulated miRNAs that have been reported in the OSCC-related literature up until 20 June 2023, providing a detailed picture of the literature of that time on miRNA expression patterns in OSCC and served as starting data pool for the conducted analyses [8]. To include novel additions of dysregulated molecules in the current literature, we conducted a new additional PubMed search, employing the same criteria. Specifically, we used the keyword combinations (“miRNAs” AND “OSCC”) and (“miRNAs” AND “expression” AND “OSCC”) and limited our search to review articles in English, published between 21 June 2023 and 10 April 2024. These searches yielded 10 and 7 results, respectively, including our previous research [8]. Among them, we selected 5 articles that (a) were specifically focused on OSCC (excluding precancerous oral pathologies and other head and neck cancers), (b) included more than 3 implicated miRNA molecules, and (c) clearly stated the expression patterns and the sample sources derived from OSCC patients that corresponded to miRNA quantification (at least including solid OSCC tumor tissue and/or saliva samples).

### 4.4. Configuration of Customized Target Gene Panels for Each Selected Stage of Oral Oncogenesis

Our genomic data were generated by leveraging our group’s representative and award-winning hamster model of sequential oral oncogenesis, which encompasses every distinct stage from oral hyperkeratosis, hyperplasia, dysplasia, and early invasion, to well-differentiated and moderately differentiated OSCC, portraying the characteristic expressional dysregulation of key implicated genes in each stage [30,31]. For the current study, we selected oral hyperplasia and dysplasia as the primary precancerous stages and ultimately the following malignant stage of early invasion, to capture the juncture where malignant transformation takes place. For each stage, two customized human gene panels were generated, depending on their expressional pattern (upregulation or downregulation) at that specific stage, according to the animal model (Figure 22). The mRNAs of the genes included in the panels were later used as pools of candidate targets for miRNA molecules based on miRNA/mRNA seed matching.

For oral hyperplasia, the upregulated gene panel consisted of 8 genes, including the *EGFR*, *ERBB2*, *JUN*, *ETS1*, *MYC*, and *MKI67* (*Ki-67*) oncogenes and the *CDKN2A* (*p16*) and *TP53* (*p53*) tumor suppressor genes. No genes were identified as downregulated in accordance with the model’s description of the stage’s genetic expressional signature; consequently, no downregulated gene panel was generated.

As to the stage of oral dysplasia, a 9-gene upregulated gene panel was created encompassing the *EGFR*, *ERBB2*, *FGFR2*, *FGFR3*, *ETS1*, *JUN*, *MYC*, and *MKI67* oncogenes, as well as the *TP53* tumor suppressor gene, which exhibits significant upregulation as well, at that stage. On the other hand, the downregulated gene panel pertaining to that stage comprised 3 genes that demonstrate significantly diminished expression, and these were the *CDKN2A* tumor suppressor gene, as well as the *NRAS* and *BCL2* oncogenes.

The malignant stage of early invasion allowed for the development of an upregulated gene panel, comprising the 5 oncogenes *EGFR*, *FGFR2*, *FGFR3*, *ETS1*, and *JUN*, as well as of a 2-gene downregulated gene panel that consisted of the *CDKN2A* tumor suppressor and the *BCL2* oncogene.

### 4.5. Analyses on miRNA/Gene Target Interactions and Selection of Stage-Specific Molecules

For miRNA–target bioinformatic identification and miRNA/target interaction analysis, we developed two miRNA regulatory networks for each selected stage of oral oncogenesis, using the “miRNet 2.0” miRNA-centric network visual analytics software (https://www.mirnet.ca/miRNet/home.xhtml, Xia Lab/McGill University, Montreal, QC, Canada, last accessed on 10 April 2024) that encompasses human epigenomic data from 13 different miRNA databases (TarBase, miRTarBase, miRecords, miR2Disease, HMDD, PhenomiR, SM2miR, PharmacomiR, EpimiR, starBase, TransmiR, ADmiRE, and TAM 2.0). One network identified all miRNA molecules that are predicted to target and, therefore, potentially regulate the expression of every gene in each stage’s upregulated gene panel and one second network comprised all the miRNA molecules that target and might influence the expression of each gene in the respective downregulated gene panel, pertaining to that particular stage.

Among the multiple miRNAs depicted in each generated network, we extracted the molecules that simultaneously target and possibly regulate the expression of more than 75% of the genes in each target-gene panel. Finally, from the subset of miRNA molecules that passed through this filtration process, we solely selected the ones that are experimentally verified to exhibit the reverse expressional dysregulation in OSCC-derived tissue and/or saliva specimens, with respect to one of their target genes. Therefore, there are two sets of results for each of the selected stages (hyperplasia, dysplasia, and early OSCC invasion), as follows: one set of downregulated and one set of upregulated miRNA molecules, that have the potential to function as stage-specific biomarkers, according to their high target scores (>75% of stage-specific target genes in each panel) and their dysregulated expressional patterns in the malignancy, which are the reverse of those of their targets (Figure 23).

## 5. Conclusions

The combination of genomic data from animal models with epigenomic data from human studies, by employing this particular methodology, transcends OSCC and holds great promise as a paradigm for precision oncology. Following the identification of stage-specific miRNA molecules in oral carcinogenesis, this approach offers a blueprint that can be extended to other cancer types with well-established genetic backgrounds and stage-specific genetic expressional signatures. This approach could accelerate the identification of key molecular players that could serve as biomarkers of high diagnostic value, while also contributing to the decoding and mapping of a pathology as complex as cancer.

## Figures and Tables

**Figure 1 ijms-25-07642-f001:**
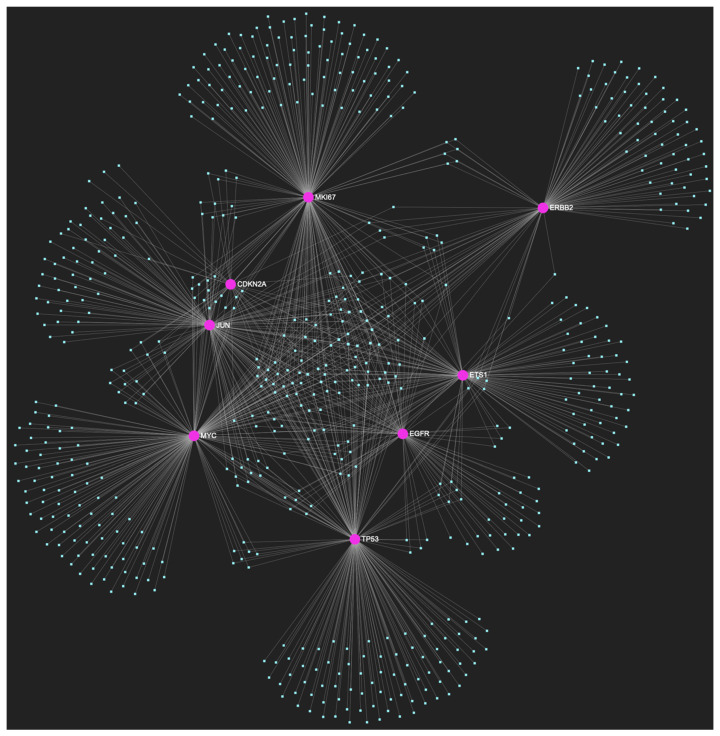
The miRNA/target interaction network, depicts the 647 miRNA molecules that are predicted to target and possibly regulate at least one of the eight genes that comprise the upregulated gene panel, developed for the stage of oral hyperplasia, according to the genomic data acquired by the hamster model of sequential oral oncogenesis. The encompassed characteristically upregulated genes for this particular stage include the *EGFR*, *ERBB2*, *JUN*, *ETS1*, *MYC*, and *MKI67* (*Ki-67*) oncogenes, as well as the *CDKN2A* (*p16*) and *TP53* (*p53*) tumor suppressor genes. The pink graphic elements illustrate the genes that comprise the panel, while the light blue dots across the illustration represent the miRNA molecules that are expected to target at least one of them, and possibly regulate its post-transcriptional expression.

**Figure 2 ijms-25-07642-f002:**
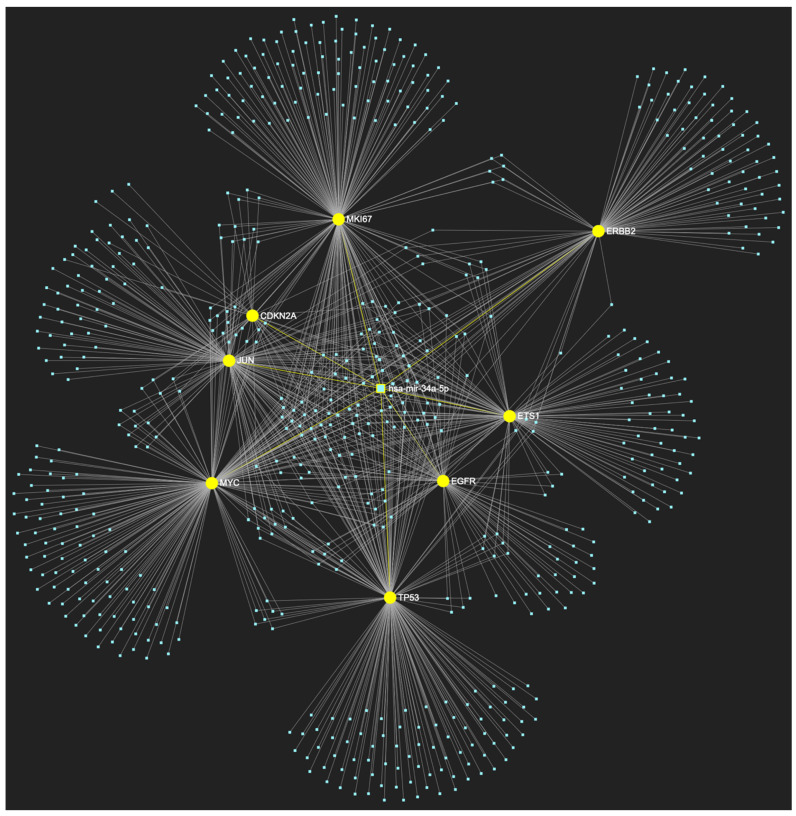
MiR-34a-5p, which has mostly been documented to exhibit decreased expression in OSCC (but has also been reported as overexpressed by a minimal number of studies), is predicted to simultaneously target and possibly regulate the expression of the total of 8 genes (target score: 100%) that are characteristically upregulated during the stage of oral hyperplasia (*EGFR*, *ERBB2*, *JUN*, *ETS1*, *MYC*, *MKI67*, *CDKN2A*, *TP53*), according to the hamster model of sequential oral oncogenesis, and comprise our stage-specific upregulated gene panel developed for that particular stage. The yellow graphic elements depict the genes that are specifically targeted by miR-34a-5p, along with their pertaining connecting nodes. The light blue dots across the illustration represent the remaining miRNA molecules that are expected to target at least one gene in this network.

**Figure 3 ijms-25-07642-f003:**
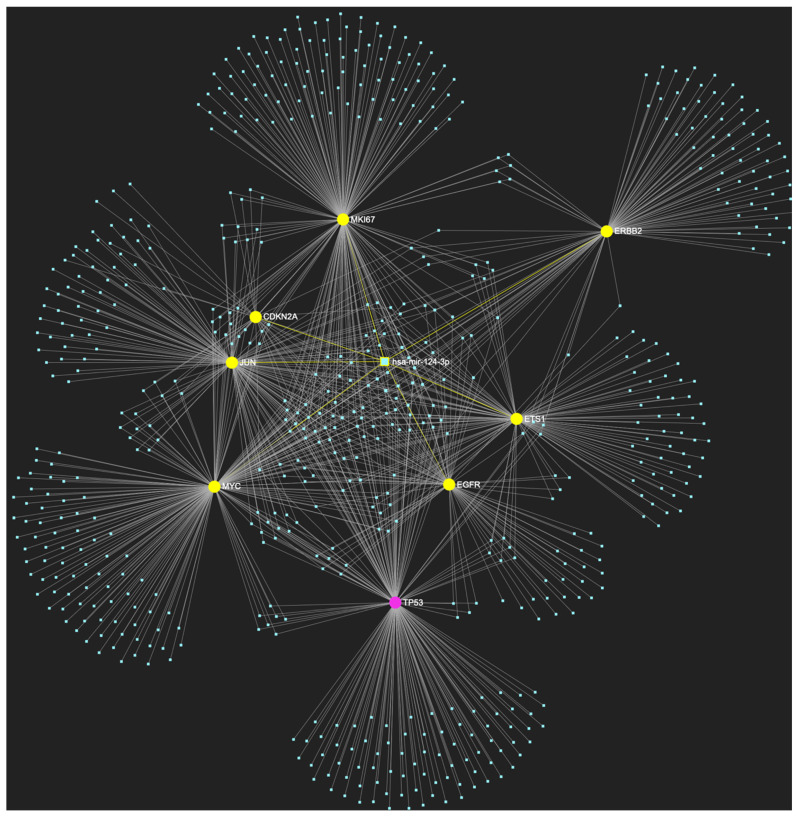
MiR-124-3p, which demonstrates significant downregulation in OSCC, is predicted to simultaneously target and possibly regulate the expression of 7 (*EGFR*, *ERBB2*, *JUN*, *ETS1*, *MYC*, *MKI67*, *CDKN2A*) out of the 8 genes that comprise the upregulated gene panel developed for the stage of oral hyperplasia, according to the hamster model of sequential oral oncogenesis, demonstrating a target score of 87.5%. The yellow graphic elements depict the genes that are specifically targeted by miR-124-3p, along with their pertaining connecting nodes. The pink elements illustrate the genes that are not targeted by this particular miRNA, while the light blue dots across the illustration represent the remaining miRNA molecules that are expected to target at least one gene in this network.

**Figure 4 ijms-25-07642-f004:**
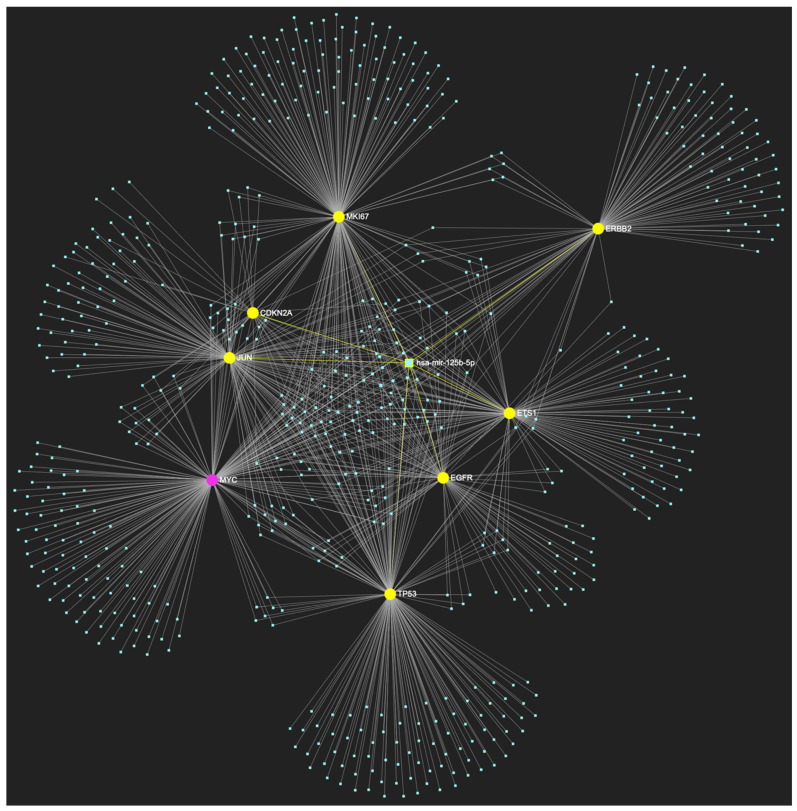
MiR-125b-5p, which is reported to be significantly under-expressed in OSCC, is predicted to simultaneously target and possibly regulate the expression of 7 (*EGFR*, *ERBB2*, *JUN*, *ETS1*, *MKI67*, *TP53*, and *CDKN2A*) out of the 8 genes that comprise the upregulated gene panel developed for the stage of oral hyperplasia, according to the hamster model of sequential oral oncogenesis (target score: 87.5%). The yellow graphic elements depict the genes that are specifically targeted by miR-125b-5p, along with their pertaining connecting nodes. The pink elements illustrate the genes that are not targeted by this particular miRNA, while the light blue dots across the illustration represent the remaining miRNA molecules that are expected to target at least one gene in this network.

**Figure 5 ijms-25-07642-f005:**
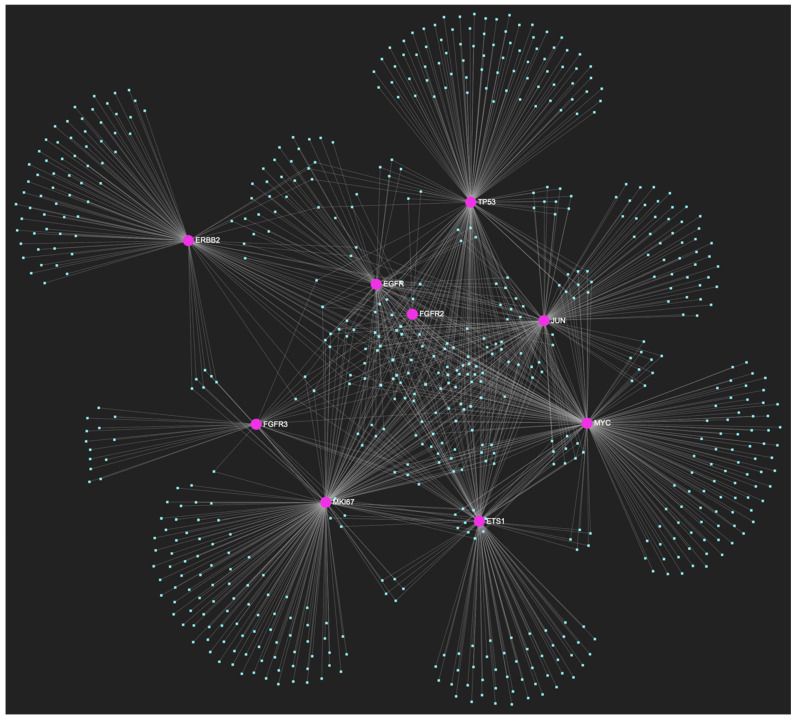
The miRNA/target interaction network, depicts the 658 miRNA molecules that are predicted to target and possibly regulate at least one of the nine genes that comprise the upregulated gene panel, developed for the stage of oral dysplasia, according to the genomic data acquired by the hamster model of sequential oral oncogenesis. The encompassed characteristically upregulated cell-cycle-regulatory genes for this particular stage include *EGFR*, *ERBB2*, *FGFR2*, *FGFR3*, *ETS1*, *MYC*, *JUN*, *TP53*, and *MKI67* (*Ki-67*). The pink graphic elements illustrate the genes that comprise the panel, while the light blue dots across the illustration represent the miRNA molecules that are expected to target at least of them, and possibly regulate its post-transcriptional expression.

**Figure 6 ijms-25-07642-f006:**
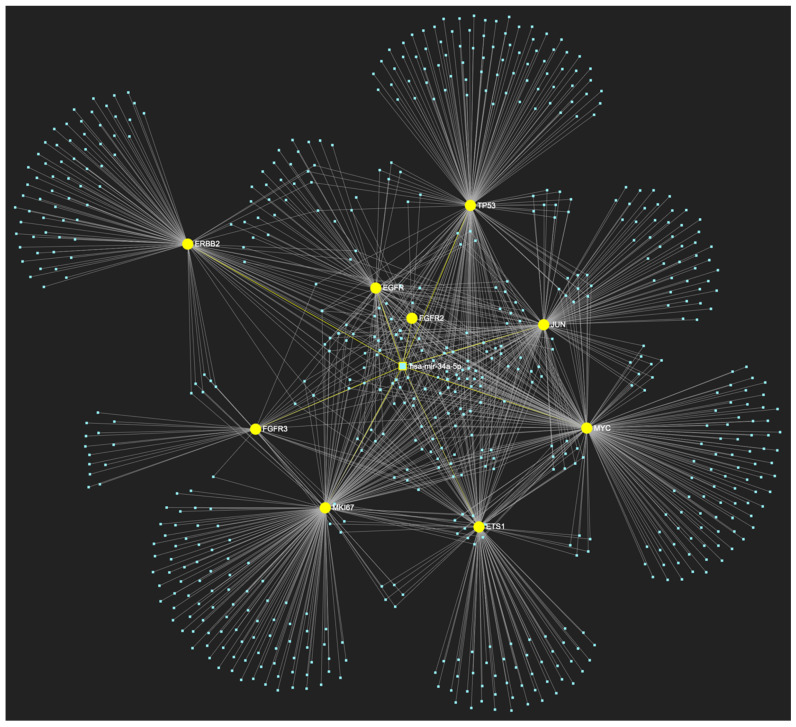
MiR-34a-5p, which is primarily reported for demonstrating reduced expression in OSCC (but has also been reported as overexpressed by a minimal number of studies), is predicted to simultaneously target and possibly regulate the expression of all of the 9 genes (target score: 100%) that are characteristically upregulated during the stage of oral dysplasia (*EGFR*, *ERBB2*, *FGFR2*, *FGFR3*, *ETS1*, *JUN*, *MYC*, *MKI67*, and *TP53*), according to the hamster model of sequential oral oncogenesis, and comprise our stage-specific upregulated gene panel developed for that particular stage. The yellow graphic elements depict the genes that are specifically targeted by miR-34a-5p, along with their pertaining connecting nodes. The light blue dots across the illustration represent the remaining miRNA molecules that are expected to target at least one gene in this network.

**Figure 7 ijms-25-07642-f007:**
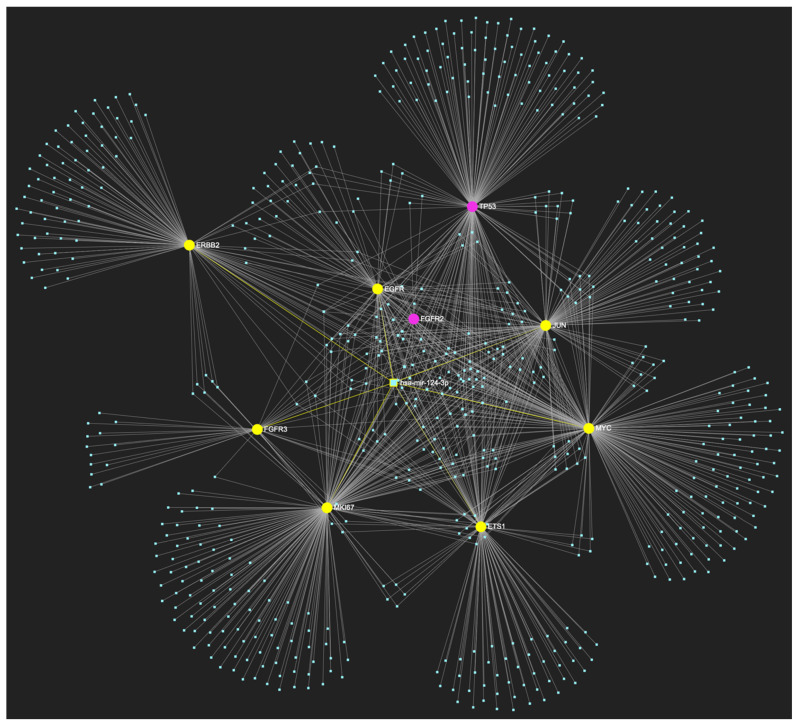
MiR-124-3p, which exhibits a significant decrease in OSCC, is predicted to simultaneously target and possibly regulate the expression of 7 (*EGFR*, *ERBB2*, *FGFR3*, *ETS1*, *MYC*, *JUN*, *MKI67*) out of the 9 genes in the upregulated gene panel developed for the stage of oral dysplasia, according to the hamster model of sequential oral oncogenesis, (target score: 77.8%). The yellow graphic elements depict the genes that are specifically targeted by miR-124-3p, along with their pertaining connecting nodes. The pink elements illustrate the genes that are not targeted by this particular miRNA, while the light blue dots across the illustration represent the remaining miRNA molecules that are expected to target at least one gene in this network.

**Figure 8 ijms-25-07642-f008:**
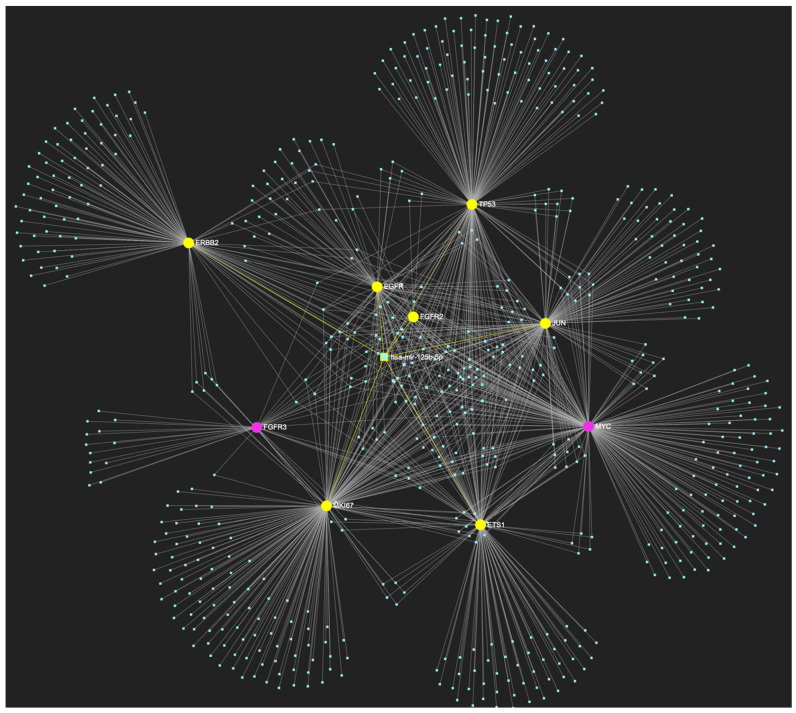
MiR-125b-5p, the levels of which are characteristically decreased in OSCC, is predicted to simultaneously target and possibly regulate the expression of 7 (*EGFR*, *ERBB2*, *FGFR2*, *ETS1*, *JUN*, *TP53*, *MKI67*) out of the 9 genes that comprise the upregulated gene panel developed for the stage of oral dysplasia, according to the hamster model of sequential oral oncogenesis, thus demonstrating a target score of 77.8%. The yellow graphic elements depict the genes that are specifically targeted by miR-125b-5p, along with their pertaining connecting nodes. The pink elements illustrate the genes that are not targeted by this particular miRNA, while the light blue dots across the illustration represent the remaining miRNA molecules that are expected to target at least one gene in this network.

**Figure 9 ijms-25-07642-f009:**
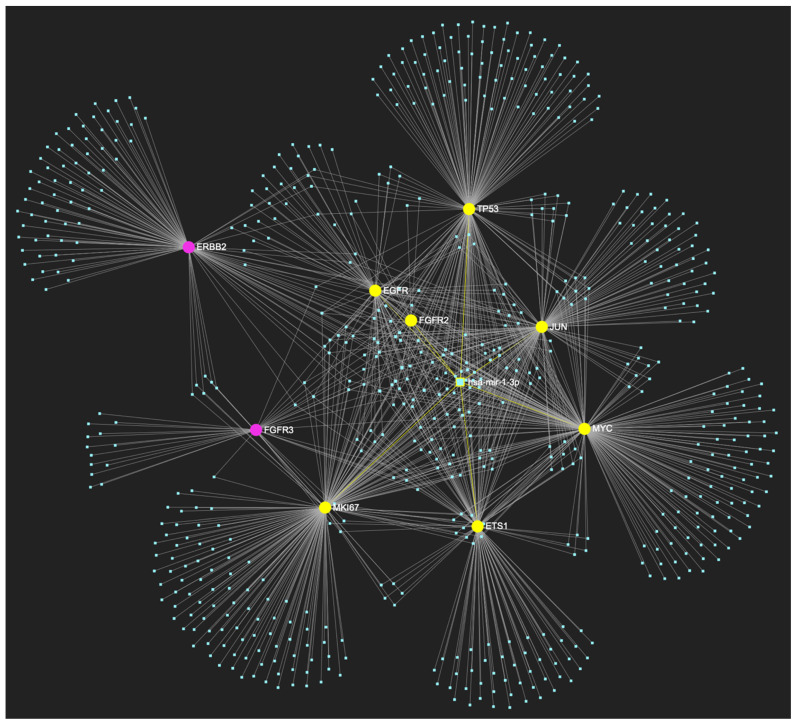
MiR-1-3p, which is significantly under-expressed in OSCC, is predicted to target and potentially regulate the expression of 7 (*EGFR*, *FGFR2*, *ETS1*, *MYC*, *JUN*, *TP53*, *MKI67*) out of the 9 genes that comprise the upregulated gene panel developed for the stage of oral dysplasia, according to the hamster model of sequential oral oncogenesis, demonstrating a target score of 77.8%. The yellow graphic elements depict the genes that are specifically targeted by miR-1-3p, along with their pertaining connecting nodes. The pink elements illustrate the genes that are not targeted by this particular miRNA, while the light blue dots across the illustration represent the remaining miRNA molecules that are expected to target at least one gene in this network.

**Figure 10 ijms-25-07642-f010:**
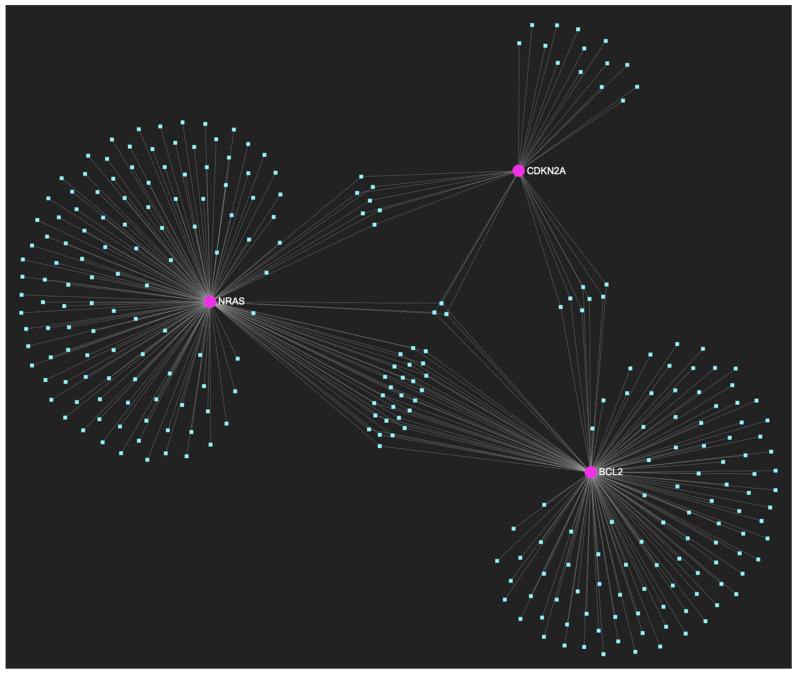
The miRNA/target interaction network illustrates the 276 miRNA molecules that are predicted to target and possibly regulate at least one out of the three genes that comprise the downregulated gene panel, developed for the precancerous stage of oral dysplasia, according to the genomic data acquired by the hamster model of sequential oral oncogenesis, comprised of the *NRAS* and *BCL2* oncogenes, as well as the *CDKN2A* tumor suppressor gene. The pink graphic elements illustrate the genes that comprise the panel, while the light blue dots across the illustration represent the miRNA molecules that are expected to target at least one of them, and possibly regulate its post-transcriptional expression.

**Figure 11 ijms-25-07642-f011:**
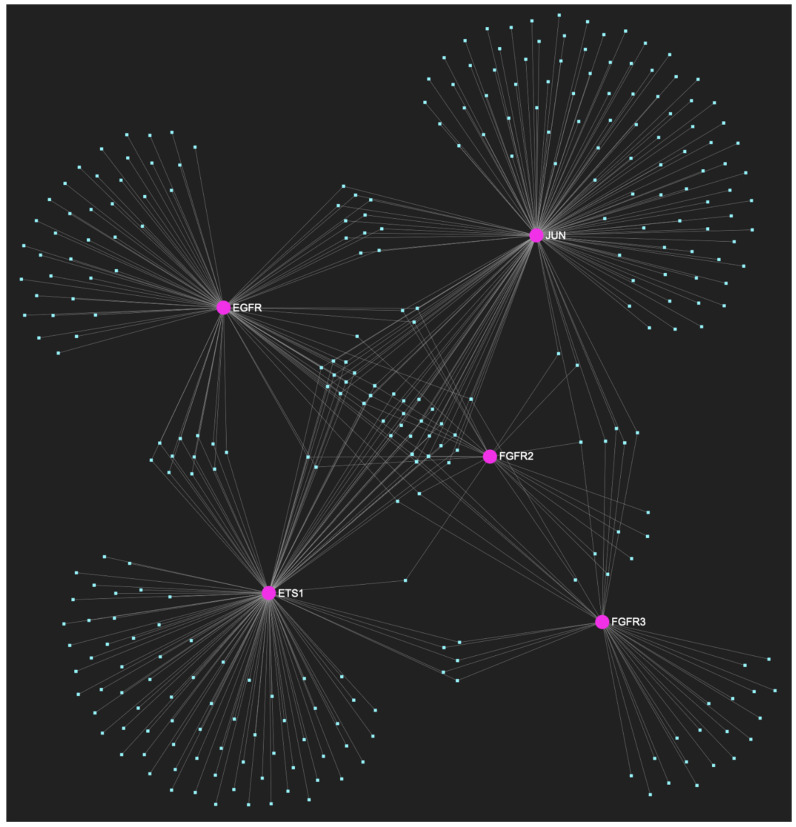
The miRNA/target interaction network, depicts the 303 miRNA molecules that are predicted to target and possibly regulate at least one of the five oncogenes (*EGFR*, *FGFR2*, *FGFR3*, *ETS1*, and *JUN*) that comprise the upregulated gene panel, developed for the initial cancerous stage of early invasion, according to the genomic data acquired by the hamster model of sequential oral oncogenesis. The pink graphic elements illustrate the genes that comprise the panel, while the light blue dots across the illustration represent the miRNA molecules that are expected to target at least of them, and possibly regulate its post-transcriptional expression.

**Figure 12 ijms-25-07642-f012:**
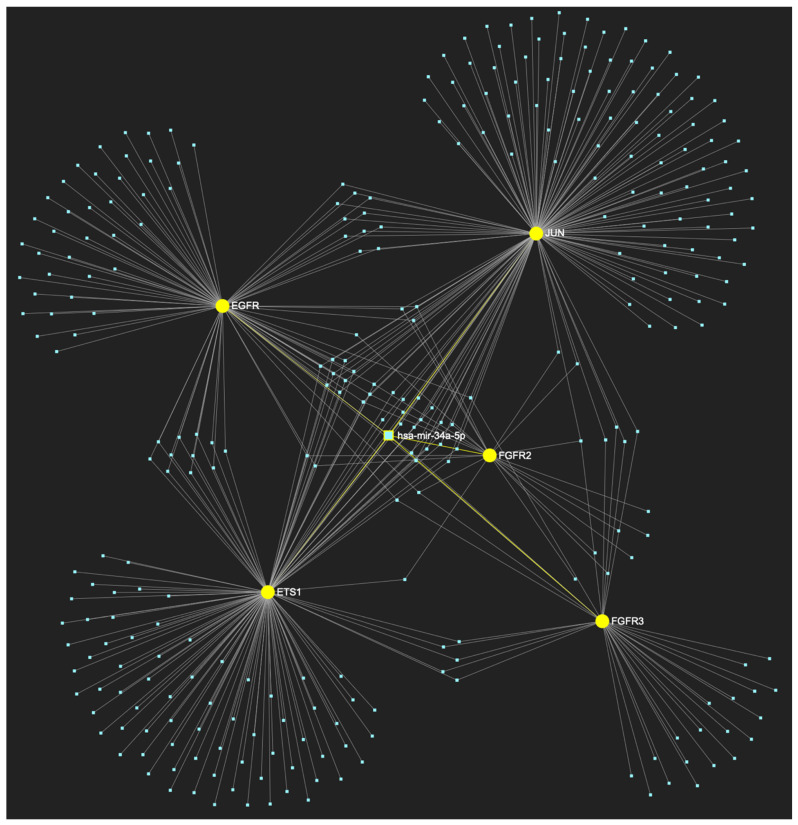
MiR-34a-5p, which is primarily reported for demonstrating reduced expression in OSCC (but has also been reported as overexpressed by a minimal number of studies), is predicted to target and possibly regulate the expression of all 5 oncogenes (target score: 100%), which are characteristically overexpressed during the initial cancerous stage of early invasion (*EGFR*, *FGFR2*, *FGFR3*, *ETS1*, and *JUN*), according to the hamster model of sequential oral oncogenesis, and comprise our stage-specific upregulated gene panel developed for that particular stage. The yellow graphic elements depict the genes that are specifically targeted by miR-34a-5p, along with their pertaining connecting nodes. The light blue dots across the illustration represent the remaining miRNA molecules that are expected to target at least one gene in this network.

**Figure 13 ijms-25-07642-f013:**
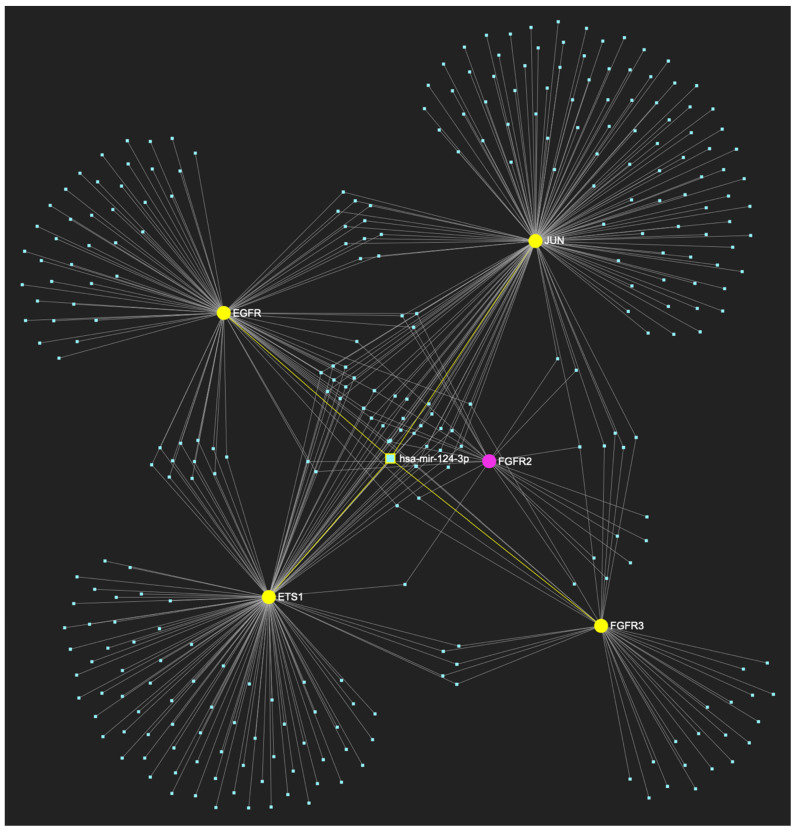
MiR-124-3p, which exhibits a significant decrease in OSCC, is predicted to simultaneously target and possibly regulate the expression of 4 (*EGFR*, *FGFR3*, *ETS1*, and *JUN*) out of 5 genes comprising the upregulated gene panel established for the initial cancerous stage of early invasion, according to the hamster model of sequential oral oncogenesis, demonstrating a “Target Score” of 80%. The yellow graphic elements depict the genes that are specifically targeted by miR-124-3p, along with their pertaining connecting nodes. The pink elements illustrate the genes that are not targeted by this particular miRNA, while the light blue dots across the illustration represent the remaining miRNA molecules that are expected to target at least one gene in this network.

**Figure 14 ijms-25-07642-f014:**
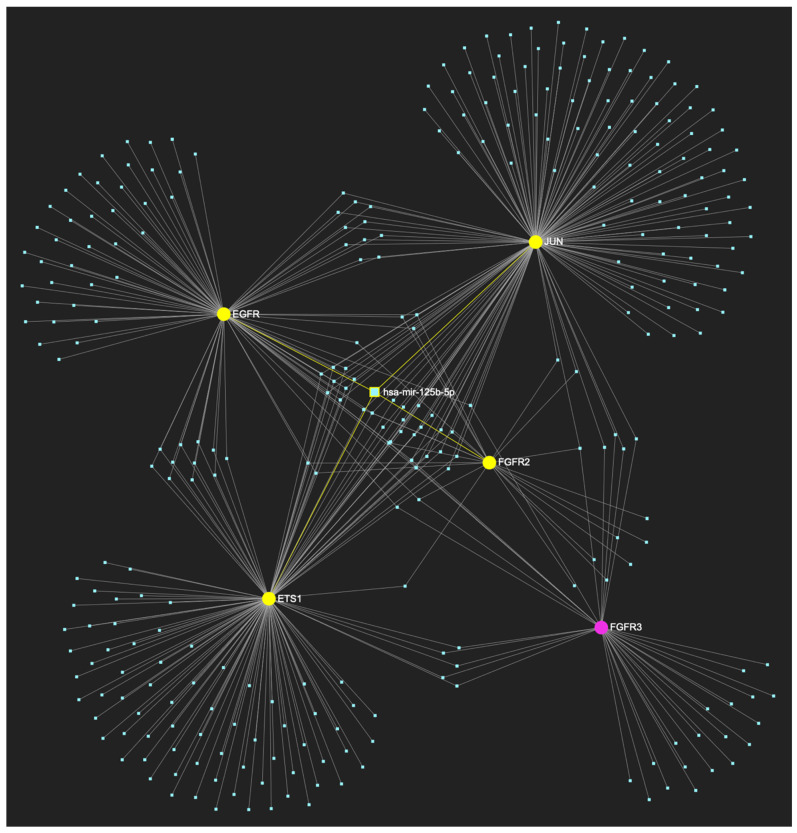
MiR-125b-5p, the levels of which are characteristically decreased in OSCC, is predicted to simultaneously target and possibly regulate the expression of 4 (*EGFR*, *FGFR2*, *ETS1*, and *JUN*) out of 5 genes that make up the upregulated gene panel developed for the initial malignant stage of early invasion, according to the hamster model of sequential oral oncogenesis, suggesting a “Target Score” of 80%. The yellow graphic elements depict the genes that are specifically targeted by miR-125b-5p, along with their pertaining connecting nodes. The pink elements illustrate the genes that are not targeted by this particular miRNA, while the light blue dots across the illustration represent the remaining miRNA molecules that are expected to target at least one gene in this network.

**Figure 15 ijms-25-07642-f015:**
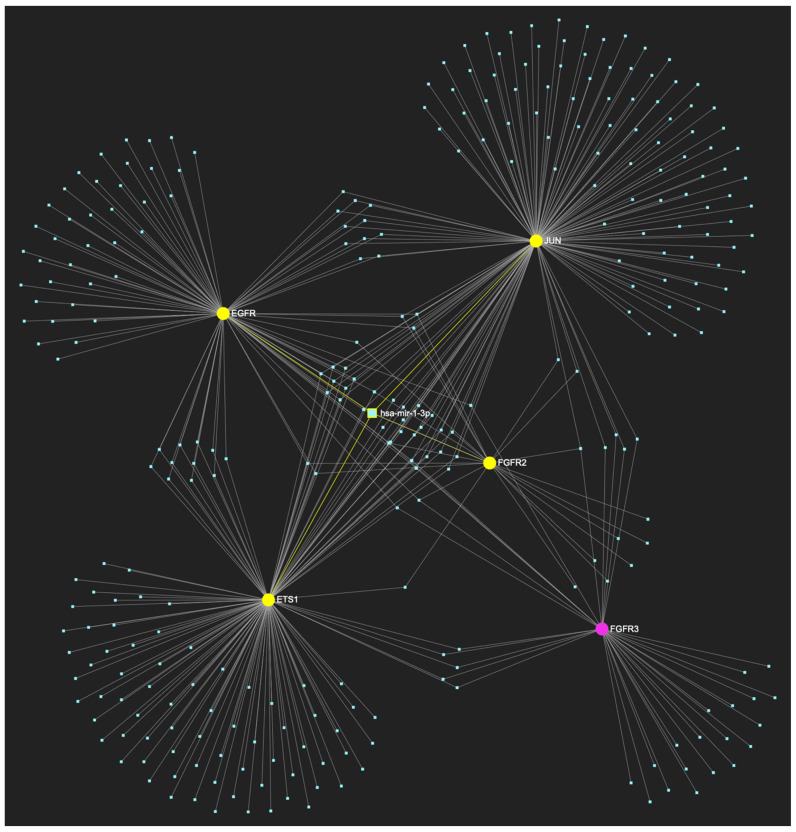
MiR-1-3p, which is significantly under-expressed in OSCC, is expected to target and potentially control the expression of 4 (*EGFR*, *FGFR2*, *ETS1*, and *JUN*) out of 5 genes that make up the upregulated gene panel developed for the initial malignant stage of early invasion, according to the hamster model of sequential oral oncogenesis, signifying a “Target Score” of 80%. The yellow graphic elements depict the genes that are specifically targeted by miR-1-3p, along with their pertaining connecting nodes. The pink elements illustrate the genes that are not targeted by this particular miRNA, while the light blue dots across the illustration represent the remaining miRNA molecules that are expected to target at least one gene in this network.

**Figure 16 ijms-25-07642-f016:**
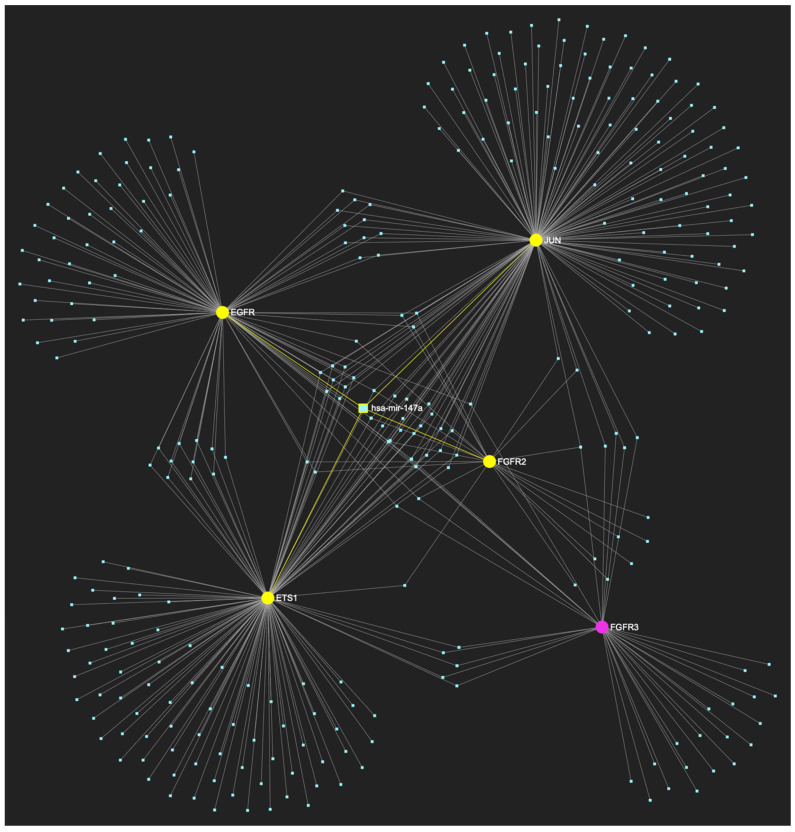
MiR-147a, which is boldly reduced in OSCC, is expected to target and possibly regulate the expression of 4 (*EGFR*, *FGFR2*, *ETS1*, and *JUN*) out of 5 genes comprising the upregulated gene panel established for the initial malignant stage of early invasion, according to the hamster model of sequential oral oncogenesis, indicating a “Target Score” of 80%. The yellow graphic elements depict the genes that are specifically targeted by miR-147a, along with their pertaining connecting nodes. The pink elements illustrate the genes that are not targeted by this particular miRNA, while the light blue dots across the illustration represent the remaining miRNA molecules that are expected to target at least one gene in this network.

**Figure 17 ijms-25-07642-f017:**
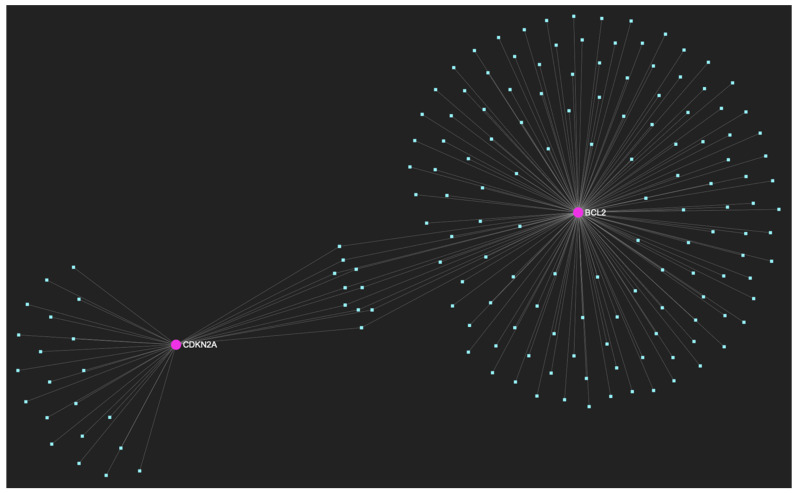
This miRNA/target interaction network demonstrates the 159 miRNA molecules that are predicted to target and possibly regulate at least one of the two genes (*CDKN2A* and *BCL2*) encompassing the downregulated gene panel developed for the initial malignant stage of early invasion, according to the genomic data acquired by the hamster model of sequential oral oncogenesis. The pink graphic elements illustrate the genes that comprise the panel, while the light blue dots across the illustration represent the miRNA molecules that are expected to target at least one of them, and possibly regulate its post-transcriptional expression.

**Figure 18 ijms-25-07642-f018:**
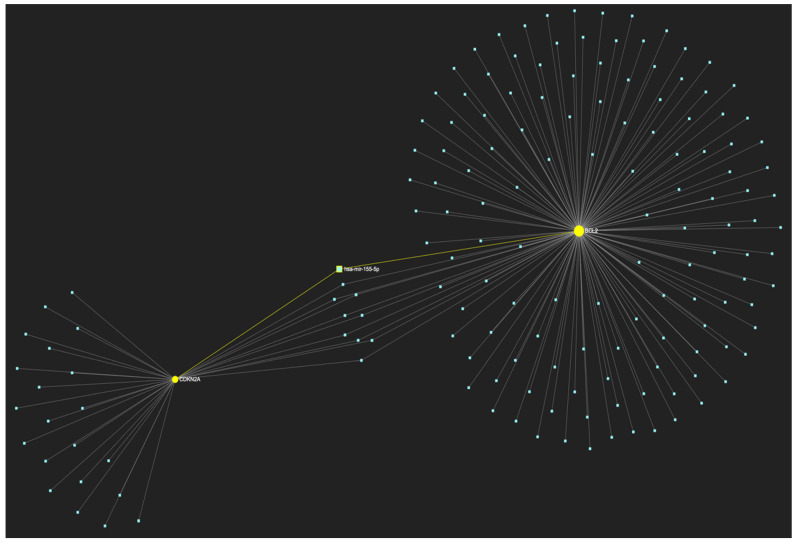
MiR-155-5p, a molecule that has been extensively studied and found to be increased in OSCC, is expected to target and possibly regulate the expression of both CDKN2A and BCL2 genes (target score: 100%), which exhibit significant downregulation during the stage of early invasion, according to the hamster model of sequential oral oncogenesis, and comprise our stage-specific upregulated gene panel developed for that particular stage. The yellow graphic elements depict the genes that are specifically targeted by miR-155-5p, along with their pertaining connecting nodes. The light blue dots across the illustration represent the remaining miRNA molecules that are expected to target at least one gene in this network.

**Figure 19 ijms-25-07642-f019:**
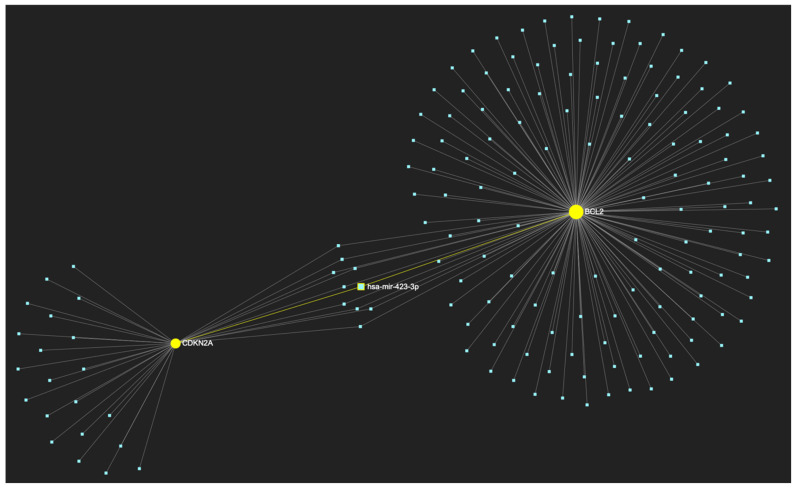
MiR-423-3p, a molecule that has been reported to demonstrate significant upregulation in OSCC, is predicted to target and possibly regulate the expression of both *CDKN2A* and *BCL2* genes (target score: 100%), which exhibit significant downregulation during the stage of early invasion according to the hamster model of sequential oral oncogenesis. The yellow graphic elements illustrate the genes that are specifically targeted by miR-423-3p, along with their pertaining connecting nodes. The light blue dots across the illustration represent the remaining miRNA molecules that are expected to target at least one gene in this network.

**Figure 20 ijms-25-07642-f020:**
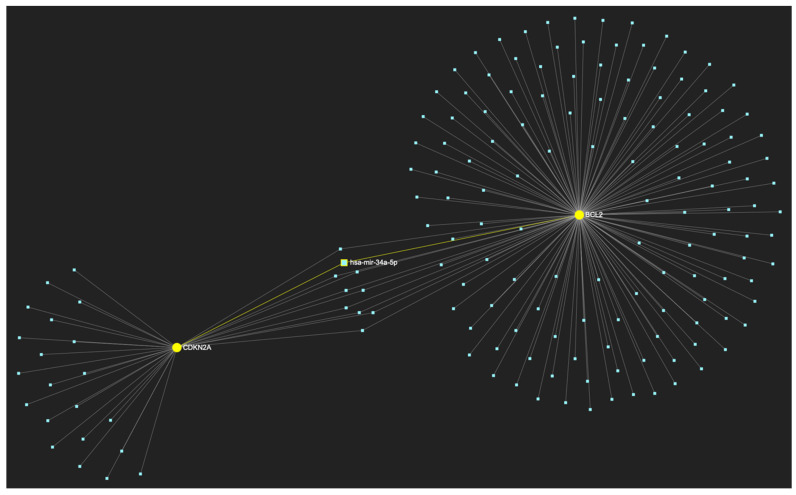
MiR-34a-5p, which has been reported as overexpressed by a minimal number of studies, in contrast with the majority that portrays it as a significantly downregulated molecule in OSCC, is predicted to target and possibly regulate the expression of both *CDKN2A* and *BCL2* genes (target score: 100%), which exhibit significant downregulation during the stage of early invasion, according to the hamster model of sequential oral oncogenesis, and comprise our stage-specific upregulated gene panel developed for that particular stage. The yellow graphic elements depict the genes that are specifically targeted by miR-34a-5p, along with their pertaining connecting nodes. The light blue dots across the illustration represent the remaining miRNA molecules that are expected to target at least one gene in this network.

**Figure 21 ijms-25-07642-f021:**
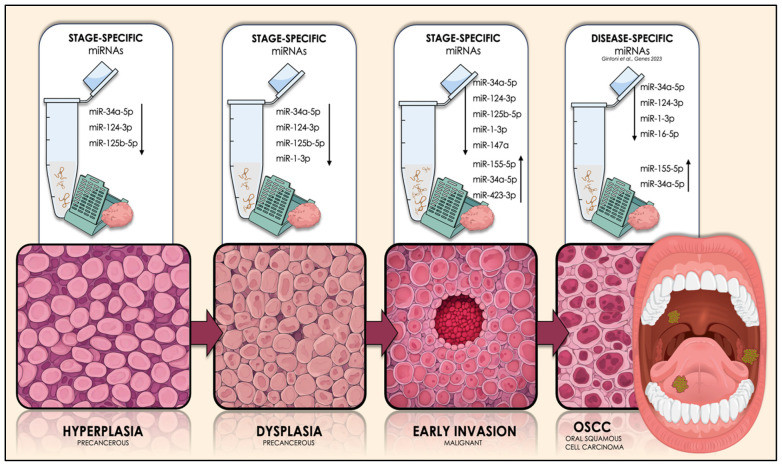
Graphical illustration of the key miRNA molecules that are expected to govern each initial stage of OSCC tumorigenesis (hyperplasia, dysplasia, and early invasion of OSCC cells), as well as the overall malignancy, by exhibiting dysregulated expression patterns, as determined by our current and preceding research. The arrows next to each miRNA subset in each stage illustrate the kind of the expressional dysregulation of those molecules in OSCC-derived tissue and/or saliva specimens. The upward arrows depict the experimentally-verified significant overexpression of the respective molecules in those biospecimens, while the downward arrows represent their significant downregulation, compared to respective normal/control samples. The dysregulation of the illustrated molecules has been strongly associated with OSCC, among over 250 other miRNAs in the available literature. The revealing of their disease-specific and stage-specific roles, alongside their stability and feasible quantification in those accessible biological materials, highlights their potential as diagnostic biomarkers that can possibly depict the histological stage of oral mucosa in real time. Their potential clinical utilization transcends the diagnosis of a suspicious oral lesion, but might also hold the key to presymptomatic detection of OSCC, during its earliest stages as a part of regular oral screening.

**Figure 22 ijms-25-07642-f022:**
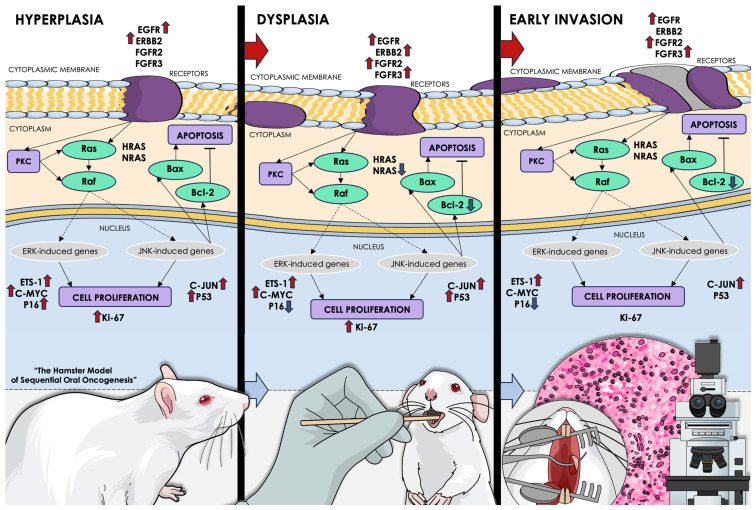
The hamster model of sequential oral oncogenesis [30,31] has been developed in order to genetically map the multistep process of OSCC tumorigenesis in Syrian hamsters, a model organism, ideal for the study of human oral pathologies. Oral carcinogenesis was chemically induced in the involved animals through the topical application of 0.5% DMBA carcinogen on the inner lining of the buccal mucosa. The developed tumors were excised after 10 weeks of carcinogen application and were subjected to pathological assessment and immunohistochemistry staining with fluorescent antibodies, for the expressional quantification of multiple OSCC-associated factors. The expression levels of the *EGFR*, *ERBB2*, *ERBB3*, *FGFR2*, *FGFR3*, *MYC*, *NRAS*, *ETS1*, *HRAS*, C-*FOS*, *JUN*, and *MKI67* oncogenes, the apoptosis markers *BAX* and *BCL2*, and the tumor suppressor genes *TP53* and *CDKN2A* (*P16*), were assessed in each stage in order to depict its respective dysregulated genetic signature. The pertained sequential stages of oral oncogenesis included normal mucosa, hyperkeratosis, hyperplasia, dysplasia, early invasion, well-differentiated, and moderately differentiated OSCC. For this study, we selected oral hyperplasia and dysplasia as the primary precancerous stages and ultimately the following malignant stage of early invasion, in order to capture the juncture where malignant transformation takes place, which also encompasses the “stage 0” of carcinoma in situ. In its upper part, this figure illustrates the signal transduction pathways governing oral oncogenesis, while the expressional dysregulation of the examined variables is displayed by arrows pointing upwards or downwards in each histological stage (hyperplasia, dysplasia, and early invasion). The red upward arrows indicate a significant increase in expression during those stages, as compared to normal oral mucosa (for the precancerous hyperplasia and dysplasia) or compared to the mean expression of these factors in normal mucosa and precancerous stages (for the malignant stage of early invasion). The downward blue arrows illustrate the significant downregulation of the studied variables in the depicted stages. Lastly, in the lower portion, a brief depiction of the methodology employed in this experimental animal system is provided.

**Figure 23 ijms-25-07642-f023:**
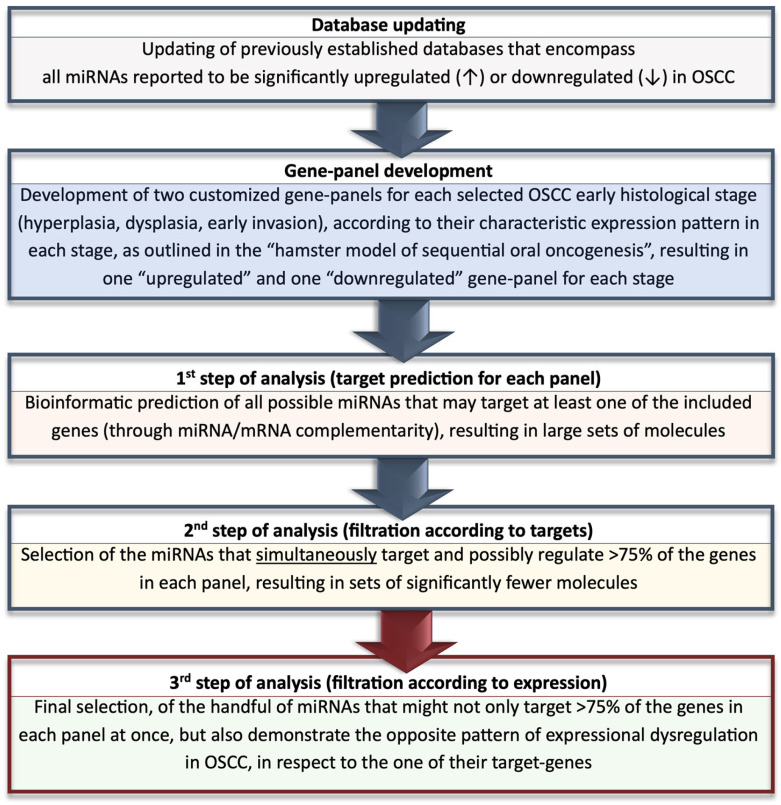
Concise flowchart providing an end-to-end overview of the methodology employed during the present research. This flowchart delivers a brief description of the sequential steps involved in our in silico analyses and the criteria used for the bioinformatic filtering of the miRNA results of each phase, in order for the most specific molecules for each elected stage of OSCC tumorigenesis (oral hyperplasia, dysplasia, and early invasion of OSCC cells) to be yielded. The final stage-specific molecules were selected based on a combination of their target genes and target scores for each panel, as well as the patterns of their distinctive expressional dysregulation in the disease.

**Table 1 ijms-25-07642-t001:** MiRNA molecules that have been experimentally documented to demonstrate significantly upregulated and downregulated expression levels in OSCC-derived tissue and/or saliva specimens, compared to normal respective biosamples and were included in the relevant literature that met the criteria of our search in the time period between 20 June 2023, and 11 April 2024. The 7 encompassed miRNA molecules were added to the previously developed database of 106 upregulated miRNAs in OSCC, as well as 10 molecules in the respective developed database of 133 downregulated miRNAs in the created OSCC database [8].

**Upregulated miRNAs in OSCC (Tissue and/or Saliva Specimens)**	
**↑ miRNA**	**Sample Source**
miR-1307-5p	Saliva (exosomes) [26]
miR-193b-3p	Saliva [26]
miR-19a-3p	Tissue [28]
miR-200c-3p	Tissue, cell-lines [6]
miR-23a-3p	Tissue, cell-lines [6]
miR-345-5p	Tissue, saliva [26,27]
miR-378a	Tissue, cell-lines [6]
**Downregulated miRNAs in OSCC (Tissue and/or Saliva Specimens)**	
**↓ miRNA**	**Sample Source**
let-7g-5p	Tissue [28]
miR-133b	Tissue [27]
miR-140-5p	Saliva (exosomes) [26]
miR-143-5p	Saliva (exosomes) [26]
miR-15a-5p	Saliva [26]
miR-16-1-3p	Saliva [26]
miR-30c-5p	Saliva [26]
miR-363-3p	Tissue [28]
miR-3928	Saliva [26]
miR-424-3p	Saliva [26]

**Table 2 ijms-25-07642-t002:** Overview of the results obtained from the miRNA/target interaction analysis and stage-specific miRNA selection process for the stage of oral hyperplasia. MiR-34a-5p simultaneously targets 100% (8/8) of characteristically upregulated genes for this particular stage, while miR-124-3p and miR-125b-5p target and possibly regulate 87.5% (7/8) of the developed panel’s target genes. All three miRNAs exhibit significant downregulation in tissue and/or saliva specimens derived from OSCC patients, compared to non-OSCC samples.

miRNA	Reported Expression in OSCC	Predicted Target Genes Upregulated in Oral Hyperplasia (8)	Target Score
hsa-miR-34a-5p	**↓**	*EGFR*, *ERBB2*, *JUN*, *ETS1*, *MYC*, *MKI67*, *TP53*, *CDKN2A*	**8/8** (100%)
hsa-miR-124-3p	**↓**	*EGFR*, *ERBB2*, *JUN*, *ETS1*, *MYC*, *MKI67*, *CDKN2A*	**7/8** (87.5%)
hsa-miR-125b-5p	**↓**	*EGFR*, *ERBB2*, *JUN*, *ETS1*, *MKI67*, *TP53*, *CDKN2A*	**7/8** (87.5%)

**Table 3 ijms-25-07642-t003:** Overview of the results obtained from the miRNA/target interaction analysis and stage-specific miRNA selection process for the precancerous stage of oral dysplasia. MiR-34a-5p simultaneously targets 100% (9/9) of characteristically upregulated genes for this particular stage, while miR-124-3p, miR-125b-5p, and miR-1-3p target and possibly regulate 77.8% (7/9) of the developed panel’s target genes. All three miRNAs exhibit significant downregulation in tissue and/or saliva specimens derived from OSCC patients, compared to non-OSCC samples.

miRNA	Reported Expression in OSCC	Predicted Target Genes Upregulated in Oral Dysplasia (9)	Target Score
hsa-miR-34a-5p	**↓**	*EGFR*, *ERBB2*, *FGFR2*, *FGFR3*, *ETS1*, *MYC*, *JUN*, *TP53*, *MKI67*	**9/9** (100%)
hsa-miR-124-3p	**↓**	*EGFR*, *ERBB2*, *FGFR3*, *ETS1*, *MYC*, *JUN*, *MKI67*	**7/9** (77.8%)
hsa-miR-125b-5p	**↓**	*EGFR*, *ERBB2*, *FGFR2*, *ETS1*, *JUN*, *TP53*, *MKI67*	**7/9** (77.8%)
hsa-miR-1-3p	**↓**	*EGFR*, *FGFR2*, *ETS1*, *MYC*, *JUN*, *TP53*, *MKI67*	**7/9** (77.8%)

**Table 4 ijms-25-07642-t004:** Overview of the results obtained from the miRNA/target interaction analysis and stage-specific miRNA selection process for both panels. Regarding the upregulated gene panel, which characterizes the initial cancerous stage of early invasion for OSCC, miR-34a-5p simultaneously targets 100% (5/5) of the significantly upregulated genes for this particular stage, while miR-124-3p, miR-1-3p, miR-125b-5p, and miR-147a target and possibly regulate 80% (4/5) of the developed panel’s target genes, in two different combinations. The total of these five miRNAs exhibits significantly decreased levels in tissue and/or saliva biosamples derived from OSCC patients, compared to normal/control specimens. As to the downregulated gene panel, reflecting the first malignant stage of OSCC oncogenesis, namely “early invasion”, only 3 miRNA molecules (miR-155-5p, miR-423-3p, and miR-34a-5p) simultaneously target and potentially regulate the expression of both CDKN2A and BCL2 genes, while demonstrating increased expression levels in OSCC tissue and/or saliva specimens.

**miRNA**	**Reported Expression** **in OSCC**	**Predicted Target Genes Upregulated in OSCC Early Invasion (5)**	**Target Score**
hsa-miR-34a-5p	**↓** (mostly)	*EGFR*, *FGFR2*, *FGFR3*, *ETS1*, *JUN*	**5/5** (100%)
hsa-miR-124-3p	**↓**	*EGFR*, *FGFR3*, *ETS1*, *JUN*	**4/5** (80%)
hsa-miR-125b-5p	**↓**	*EGFR*, *FGFR2*, *ETS1*, *JUN*	**4/5** (80%)
hsa-miR-1-3p	**↓**	*EGFR*, *FGFR2*, *ETS1*, *JUN*	**4/5** (80%)
hsa-miR-147a	**↓**	*EGFR*, *FGFR2*, *ETS1*, *JUN*	**4/5** (80%)
**miRNA**	**Reported Expression in OSCC**	**Predicted Target Genes Downregulated in OSCC Early Invasion (2)**	**Target Score**
hsa-miR-155-5p	**↑**	*CDKN2A*, *BCL2*	**2/2** (100%)
hsa-miR-423-3p	**↑**	*CDKN2A*, *BCL2*	**2/2** (100%)
hsa-miR-34a-5p	**↑** (rarely)	*CDKN2A*, *BCL2*	**2/2** (100%)

**Table 5 ijms-25-07642-t005:** Inclusive stage-specific results for all selected early stages of OSCC oncogenesis, as well as disease-specific findings for overall OSCC. Early invasion, dysplasia, and hyperplasia are characterized by the low expression of three common microRNAs (miR-34a-5p, miR124-3p, and miR-125b-5p). The precancerous dysplasia stage is distinguished from hyperplasia by the under-expressed miR-1-3p molecule. Lastly, the malignant stage of early invasion is distinguished from the two preceding precancerous stages, by the following four miRNAs: the downregulated miR-147a, as well as miR-155-5p and miR-423-3p, whose expression is significantly elevated in OSCC, but also miR-34a-5p, which has been found to be overexpressed in a restricted number of studies. Ultimately, 4 of the 5 most disease-specific miRNAs (miR-155-5p, miR-34a-5p, miR-124-3p, and miR-1-3p miR-16-5p) for overall OSCC, revealed in our previous study, are among the most specific results regarding each sequential stage of oncogenesis. More specifically, miR-34a-5p and miR-124-3p (downregulated) are found to coexist in all three selected early stages, while miR-1-3p is introduced during the stages of dysplasia and early invasion. Finally, the upregulated miR-155-5p, stands among the dysregulated miRNAs that particularly distinguish the cancerous stage of early invasion.

Oral Hyperplasia		Oral Dysplasia		Early Invasion		OSCC (Gintoni et al. 2023) [8]	
miRNA	Expression	miRNA	Expression	miRNA	Expression	miRNA	Expression
miR-34a-5p	**↓**	miR-34a-5p	**↓**	miR-34a-5p	**↓** (mostly)	miR-34a-5p	**↓**
miR-124-3p	**↓**	miR-124-3p	**↓**	miR-124-3p	**↓**	miR-124-3p	**↓**
miR-125b-5p	**↓**	miR-125b-5p	**↓**	miR-125b-5p	**↓**	miR-1-3p	**↓**
		hsa-miR-1-3p	**↓**	miR-1-3p	**↓**	miR-16-5p	**↓**
				miR-147a	**↓**	miR-155-5p	**↑**
				miR-155-5p	**↑**	miR-34a-5p	**↑** (rarely)
				miR-423-3p	**↑**		
				miR-34a-5p	**↑** (rarely)		

## Data Availability

The original contributions presented in this study are included in the article. Please direct any additional inquiries to the corresponding author.

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
