# Peer review of "Identification of Stage-Specific microRNAs that Govern the Early Stages of Sequential Oral Oncogenesis by Strategically Bridging Human Genetics with Epigenetics and Utilizing an Animal Model"

_ijms, 2024, doi:10.3390/ijms25147642_

Round 1

Reviewer 1 Report

Comments and Suggestions for Authors

The work “Identification of stage-specific microRNAs that govern the early stages of sequential oral oncogenesis by strategically bridging genetics and epigenetics” is a meaningful try for early and precise diagnosis of oral squamous cell carcinoma (OSCC). However, the data are not very convincing, for no cells or in vivo assays were conducted to validate their conclusion obtained bioinformatically.

Below are some confusing points.

1.     What’s the innovation of this paper compared with reference 8? They seem quite similar in introduction, methods and result, except that the identified miRNA biomarkers are not the same.

In line 87-90, the authors stated that “Therefore, significant challenges occur as to the interpretation of experimental data, but mainly the identification of the few truly disease-specific molecules that can be reliably utilized as biomarkers for OSCC diagnosis, especially in its early asymptomatic stages (8)”, in reference 8, they also reported some biomarkers for early diagnosis of OSCC.

2.     In line 106-110, “Building upon our prior research on OSCC, we utilized experimental epigenomic data on miRNA expression in humans and genomic expression data constituting the genetic signature of each early OSCC stage, spanning from hyperplasia and precancerous dysplasia to the early invasion of malignant OSCC cells. We based our research on the findings of the award-winning experimental hamster model of sequential oral oncogenesis, previously developed by our group (30,31)”

It’s confusing, what data are used in the work? The authors mentioned both human and hamster…

3.     The presentation of figures seems lack of logic, plz reconsider.

Author Response

We appreciate the time and effort the Reviewers have dedicated to providing their valuable feedback on our manuscript. We have been able to incorporate changes to reflect most of the suggestions provided by the Reviewers.

Reviewer 1

Overall Comments: The work “Identification of stage-specific microRNAs that govern the early stages of sequential oral oncogenesis by strategically bridging genetics and epigenetics” is a meaningful try for early and precise diagnosis of oral squamous cell carcinoma (OSCC). However, the data are not very convincing, for no cells or in vivo assays were conducted to validate their conclusion obtained bioinformatically.

Response:           We would like to thank the reviewer for the time and effort dedicated to providing their feedback and constructive criticism on our study. We completely understand that the presence of both bioinformatic and experimental methods within a study would seem as more convincing at a first glance. However, the aim of this present study is not to recreate or revalidate already performed experiments, but to provide a bioinformatic shortcut between the existing vast volume of experimental data, which have been obtained as significant by different research teams worldwide, and the documented identification of the few truly miRNA molecules that actually hold diagnostic potential particularly for the early stages of oral carcinogenesis, where timely clinical intervention is crucial but yet rare, due to their asymptomatic nature.

It is important to highlight that the presented and applied methodology utilizes already established, genetic (stage-specific gene expression) and epigenetic (OSCC miRNA expression) data that have been obtained through experimental research and has been published in the previous years, constituting the most relevant literature available in the field. As we state in the manuscript, in our previous published work, regarding overall OSCC (Reference 8), we have created two databases, which contain all miRNA molecules that have been experimentally proven to exhibit significant expressional dysregulation (up- or down-regulation) in OSCC-derived biospecimens (tissue, cell-lines, saliva and blood), so far. Those databases, being accordingly updated of course, served as an OSCC-specific data pool for filtering the results of our bioinformatic analyses and selecting the most specific molecules in this study as well.

All in all, the employed experimental data from cells or in vivo assays have been already obtained and published as statistically significant, while all 200+ miRNAs included in the aforementioned data-pool have already been proposed as potential biomarkers for the disease in the literature. Therefore, there is no need for us to recreate the experiments or revalidate them. The point of our present work is to bioinformatically filter those numerous significant results and identify the most specific molecules to serve as the most specific miRNA biomarkers for the early detection of OSCC, based on both their target OSCC-related dysregulated genes and also the reverse interplay between miRNA and target-gene levels, which is exclusively backed up by experimental results in the present study. The part of in-silico analysis solely regards miRNA/target interactions as to target prediction, which is also based on experimentally obtained, known sequences of the miRNAs and their respective mRNA targets.

Comment 1: What’s the innovation of this paper compared with reference 8? They seem quite similar in introduction, methods and result, except that the identified miRNA biomarkers are not the same.

Response:           We thank the Reviewer for pointing this out. Indeed, the two papers might seem similar in some sections, however they are substantially different., It is reasonable that in the Introduction section there are similarities regarding the general landscape of the disease, which is OSCC in both cases, the role of miRNAs as promising diagnostic tools and finally the vast number of molecules that have been detected to exhibit dysregulation (up- or down-regulation) in OSCC-obtained biospecimens, leading to the problem of “too many results” with lack of interpretation and documented identification of the most suitable molecules for liquid biopsy. This is because this particular “obstacle” was the stepping stone for the development of this combinatory in-silico methodology, presented in both studies, but for answering completely different “questions” and employing different variables in each one of them.

In our previous paper (reference 8), where this methodology was first introduced, we aimed to identify the most disease-specific miRNAs for OSCC based on their target-genes and the reverse interplay between miRNAs and their target-gene levels. In that study, OSCC was approached as a whole entity, so we created 2 panels of the most disease-specific oncogenes and tumor suppressor genes, which are over- and under-expressed wholesomely in OSCC, respectively. We also created 2 large databases comprising of all miRNA molecules that have been experimentally verified as significantly up- or down-regulated in the disease so far, according to the relevant literature. Therefore, by utilizing the overall genetic background of the disease as input and all relevant dysregulated miRNAs as variables for the filtering of the initial target-prediction results, we managed to identify the 5 molecules (among 239), which are the most specific for the disease as a whole, by bridging OSCC genetics and epigenetics. Therefore, those molecules were determined as the most significant and accurate among all the proposed biomarkers for the diagnosis of an already histologically formed OSCC tumor (regardless its size that might be small and thus hardly noticeable by the clinician or the patient).

In the present study, the “question” set to be answered is completely different. OSCC might still be the disease of interest, but in this case the aim is to identify the most specific miRNAs that particularly govern its distinct early stages, and not the disease in general. Therefore, while the same methodology is employed (as a tool for specific-miRNA identification), the variables have changed completely. The end-to-end methodology course is applied for each histological stage (hyperplasia, dysplasia, early-invasion) separately, leading to its unique set of miRNA results. In this current study, the genetic input variables were not divided in panels of overexpressed oncogenes and underexpressed tumor suppressor genes that genetically portray the malignancy as a whole. Instead, for each stage 2 mixed panels of potential target-genes were developed, according to their specific expressional dysregulation (up- or down-regulation) in each particular stage, according to the awarded experimental “hamster model of sequential oral oncogenesis”. This animal model (that is known to closely resemble human OSCC) encompasses the dysregulated genetic “signature” of each consecutive stage of oral tumorigenesis, histologically spanning from precancerous hyperkeratosis to moderately-differentiated OSCC tumors, as described in the manuscript. All the genes involved, have been strongly (experimentally) associated with the development of human OSCC, as well, but without such a clear staging system of their dysregulated expression currently available. Therefore, in this current work, the miRNAs that govern each early stage of sequential OSCC tumorigenesis, have been identified. This leads to the documented revealing of stage-specific biomarker sets that might be utilized for the accurate detection of asymptomatic precancerous hyperplasia, dysplasia and early invasion of OSCC cells within a seemingly “healthy” oral mucosa.

As we mention in Discussion, page 31, following the Reviewer’s feedback, those results are of great importance and the concept of this current study is highly innovative, since although >250 dysregulated miRNAs have been experimentally detected in OSCC and are available in the literature, the RNA extraction preceding their quantification has been performed in OSCC specimens that normally contain all preceding histological stages within the same tissue-sample. Therefore, their stage-specific role is nearly impossible to be directly portrayed in the experimental laboratory setting and has remained unseen until now. Those results are not only crucial for supporting reliable liquid-biopsy development, but also for the detailed molecular mapping of the disease itself, which can only come from the strategic bridging of key experimentally-acquired data.

Comment 2: In line 106-110, “Building upon our prior research on OSCC, we utilized experimental epigenomic data on miRNA expression in humans and genomic expression data constituting the genetic signature of each early OSCC stage, spanning from hyperplasia and precancerous dysplasia to the early invasion of malignant OSCC cells. We based our research on the findings of the award-winning experimental hamster model of sequential oral oncogenesis, previously developed by our group (30,31). It’s confusing, what data are used in the work? The authors mentioned both human and hamster….

Response: The current study constitutes an in-silico bridging of genetic (gene-expression) and epigenetic (miRNA-expression) data. The genetic data incorporated in this current study, as thoroughly presented within the manuscript, include the genetic expressional “signature” (characteristically over- and under-expressed human genes) of each selected early histological stage OSCC (precancerous hyperplasia, dysplasia and malignant early invasion), according to the experimental “hamster model of sequential oral oncogenesis”. This animal system encompasses the dysregulated genetic “signature” of each consecutive stage of oral tumorigenesis, histologically spanning from precancerous hyperkeratosis to moderately-differentiated OSCC tumors, as described in the manuscript. All genes involved in the animal model, have been strongly (experimentally) associated with human OSCC pathogenesis and are among the most important OSCC-related genes. However, this highly representative animal model was utilized because such a clear staging system of dysregulated gene-expression in the consecutive preceding histological stages of oral mucosa is currently unavailable.

Additionally, as stated in the manuscript, the golden Syrian hamster (Mesocricetus auratus) is of the best and most representative model organisms for the study of human oral diseases, and OSCC in particular, due to the respective physiological and molecular similarities to humans. The “hamster model of sequential oral oncogenesis” has been internationally honored and awarded for its crucial contribution in oral cancer research.

Following the Reviewer’s comment, we clarified in the Methodology (section 2.2., page 4) that the factors studied in the animal model were previously associated with human OSCC pathogenesis, and regarding the scope of that combinatory approach we added the paragraph: “It is crucial… challenging”.

Comment 3: The presentation of figures seems lack of logic, plz reconsider.

Response:           We thank the Reviewer raising this concern. However, all figures (except from fig.1 and fig.22), illustrate the results of the bioinformatic miRNA/target interaction analyses employed for the purposes of this study, in the form of miRNA/mRNA regulatory networks. As thoroughly explained in each caption, Figures 2, 6, 11, 12 and 18 depict the results of the 1st phase of the analyses, and therefore the large number miRNA molecules that are predicted to target at least one of the genes comprising each panel (for each stage), as well as a quick illustration of each gene panel. The rest miRNA/target interaction networks are the documentation of our results in respect to the genes of each panel that are predicted to be simultaneously targeted by each miRNA, which successfully passed through all the filtering steps of the analysis and was yielded as stage specific for the stages of hyperplasia, dysplasia and early invasion. Additionally, in the caption of each network, the illustration elements are carefully explained:

  1. e.g. “The pink graphic elements illustrate the genes that comprise the panel, while the light blue dots across the illustration represent the miRNA molecules that are expected to target at least one of them, and possibly regulate its post-transcriptional expression” (Figure 2).
  2. e.g. “The yellow graphic elements depict the genes that are specifically targeted by miR-147a, along with their pertaining connecting nodes. The pink elements illustrate the genes that are not targeted by this particular miRNA, while the light blue dots across the illustration represent the remaining miRNA molecules that are expected to target at least one gene in this network” (Figure 17).

As to Figure 1, the genomic data obtained by the hamster model of sequential oral oncogenesis are portrayed in terms of gene expression in each stage (upregulation or downregulation) indicated by arrows, as well as of the involved affected intracellular signal transduction pathways within the cytoplasmic membrane to the cytoplasm and nucleus. In the bottom part of the figure, the step-by-step experimental procedure that took place on the laboratory animals is depicted, which although simply illustrated, indeed lacks of the respective explanation in the  caption, which was added following the Reviewer’s useful feedback (“Oral carcinogenesis… factors”).

Figure 22 is a graphical illustration of the key miRNA molecules that are expected to govern each selected early stage of OSCC tumorigenesis (hyperplasia, dysplasia, and early invasion of OSCC cells), as well as the overall malignancy, by exhibiting dysregulated expression patterns, as determined by our current and preceding research. Additionally, to the caption, graphics are highly specific as to the histological stage, miRNA code name and number of molecules, expression patterns, as well as the biological material, in which those expressional patterns have been experimentally detected, according to the most relevant literature.

Reviewer 2 Report

Comments and Suggestions for Authors

In the presented manuscript the Authors examined the identification of stage-specific microRNAs that govern early stages of sequential oral oncogenesis. This issue is crucial and interesting. However, I have a few comments.

1. I suggest adding to the title that the analysis concerns an animal model.

2. The article is well-written. The aim of the study is very well justified. The presented results and discussion are very interesting.

3. I suggest adding a flow chart to better understand the methodology used by the Authors.

4. Data regarding the versions of the databases used were not presented. They were also not included in the citations. Most database has instructions on its website on how to prepare a citation.

5. Gene names should be presented in italics (Figure 1-21 – descriptions).

6. The sentence in lines 298-300 is incomprehensible.

7. Citations should be placed in square brackets.

8. The bibliography section should be revised by the Instructions to Authors.

Author Response

We appreciate the time and effort the Reviewers have dedicated to providing their valuable feedback on our manuscript.

Reviewer 2

Overall Comments: In the presented manuscript the Authors examined the identification of stage-specific microRNAs that govern early stages of sequential oral oncogenesis. This issue is crucial and interesting. However, I have a few comments.

Response: We are thankful for the Reviewer’s positive feedback and insightful comments on this manuscript. We have been able to incorporate changes to reflect most of the suggestions provided by the Reviewer.

Comment 1:       I suggest adding to the title that the analysis concerns an animal model.

Response:           We would like to thank the Reviewer for their suggestion. Although this particular study solely concerns humans, since both genetic (involved genes) and epigenetic input (miRNA dysregulation patterns) is human-centered, the animal model, was used as a valuable tool to accurately depict the dysregulation patterns in each histological stage of OSCC and not employed to determine their involvement in the pathogenesis of the disease, which has already been extensively and boldly confirmed in humans. Therefore, following the Reviewer’s suggestion we modified the title accordingly: "Identification of stage-specific microRNAs that govern the early stages of sequential oral oncogenesis by strategically bridging human genetics with epigenetics and utilizing an animal model”.

Comment 2: The article is well-written. The aim of the study is very well justified. The presented results

and discussion are very interesting.

Response: We would like to deeply thank the Reviewer, as we greatly value their positive comments regarding our work.

Comment 3: I suggest adding a flow chart to better understand the methodology used by the Authors.

Response:  We appreciate the Reviewer’s comment for bringing this to our attention. We acknowledge that while the methodology has been described in detail, it is certainly challenging. Therefore, we agree that a flowchart would be beneficial in simplifying it and allowing for a quicker comprehension at a glance.  Hence, we have included a flowchart at the end of the manuscript, as per their consultation (Figure 23).

Comment 4: Data regarding the versions of the databases used were not presented. They were also

not included in the citations. Most database has instructions on its website on how to prepare a citation.

Response: We thank the Reviewer for pointing this out. Nevertheless, as mentioned in the manuscript, we did not utilize any online databases. The integrated datasets were manually constructed in our earlier study (Gintoni et al. 2023 – reference 8), and subsequently updated for the present research.  Thus, the databases mentioned in this work are cited, and since the corresponding paper is open access, anyone can review them in the original article. The outcomes of the updating process are included in this manuscript in the form of a table (Table 1).

Comment 5: Gene names should be presented in italics (Figure 1-21 – descriptions).

Response:           We appreciate the Reviewer’s observation. We certainly concur with this and have corrected the gene names’ font throughout all of the figure descriptions.

Comment 6: The sentence in lines 298-300 is incomprehensible.

Response: We thank the Reviewer for pointing this out. Inadvertently, a portion of the sentence was skipped over, prompting its addition as a correction, following their feedback (page 8, section 3.2: “The bioinformatic strategy… molecules”). Additionally, all opening sentences of relevant sections 3.3 and 3.4 have been slightly paraphrased, in order to be more comprehensible and reader-friendly.

Comment 7: Citations should be placed in square brackets.

Response: We appreciate the Reviewer’s input and we have carefully integrated it throughout the entire manuscript, replacing the parentheses with brackets, as indicated.

Comment 8: The bibliography section should be revised by the Instructions to Authors.

Response: Following the Reviewer’s feedback, the bibliography section has been thoroughly revised and adjusted in accordance with the guidelines provided in the "Instructions to Authors" section.

Round 2

Reviewer 1 Report

Comments and Suggestions for Authors

The data are not very convincing, for no cells or in vivo assays were conducted to validate their conclusion obtained bioinformatically. The author's explanation alone is not credible. 

Author Response

Reviewer’s comment:

“The data are not very convincing, for no cells or in vivo assays were conducted to validate their conclusion obtained bioinformatically. The author's explanation alone is not credible.”

Response:

The explanation we have previously supplied to the Reviewer includes only details that have been meticulously documented in the provided manuscript.

The present research is not a theoretical bioinformatic model that needs experimental validation. On the contrary, it makes use of already available experimental miRNA-expression data and identifies the most stage-specific molecules based on their expression patterns and their OSCC-related target genes though an in-silico multistep analysis. The experimental data obtained from in vivo studies have already been published and utilized as genetic and epigenetic input in the present work, which creates a bioinformatic shortcut to the most important miRNAs from the plethora of observed dysregulated molecules. Therefore, there is no justification for duplicating every experiment that makes up the pertinent literature, nor does our research aim to do so.

Furthermore, since it is practically unfeasible to achieve precise miRNA quantification for every histological stage of OSCC in the laboratory setting, we are presenting this developed and awarded methodology tailored in a stage-specific manner.